# Analysis of localized cAMP perturbations within a tissue reveal the effects of a local, dynamic gap junction state on ERK signaling

**João Pedro Fonseca**[1‡], **Elham Aslankoohi**[2], **Andrew H. Ng**[3], **Michael Chevalier**[4‡*]

**1** Amyris Bio, Porto, Portugal, **2** Department of Electrical and Computer Engineering, University of California Santa Cruz, Santa Cruz, California, United States of America, **3** Outpace Bio, Seattle, Washington, United States of America, **4** Department of Biochemistry and Biophysics, University of California San Francisco, San Francisco, California, United States of America

‡ These authors share first authorship on this work.
* mikchevy@alumni.stanford.edu

**Data Availability Statement:** Data is available at the Dryad Digital Repository: https://doi.org/10.7272/Q6QC01RN. Code for the data processing and analysis, and the computational model is

## Abstract

Beyond natural stimuli such as growth factors and stresses, the ability to experimentally modulate at will the levels or activity of specific intracellular signaling molecule(s) in specified cells within a tissue can be a powerful tool for uncovering new regulation and tissue behaviors. Here we perturb the levels of cAMP within specific cells of an epithelial monolayer to probe the time-dynamic behavior of cell-cell communication protocols implemented by the cAMP/PKA pathway and its coupling to the ERK pathway. The time-dependent ERK responses we observe in the perturbed cells for spatially uniform cAMP perturbations (all cells) can be very different from those due to spatially localized perturbations (a few cells). Through a combination of pharmacological and genetic perturbations, signal analysis, and computational modeling, we infer how intracellular regulation and regulated cell-cell coupling each impact the intracellular ERK response in single cells. Our approach reveals how a dynamic gap junction state helps sculpt the intracellular ERK response over time in locally perturbed cells.

## Author summary

To effectively respond to various perturbations, cells within in-vivo tissues must coordinate through intimate collaboration and sharing of information. Acquiring a quantitative understanding of the underlying biochemical signaling and cell-cell communication involved may require one to test the tissue response under various types of controlled spatiotemporal perturbations. While this may be hard, if not impossible, to do for an in-vivo tissue, in-vitro models can often serve as an approximate substitute. In this work we demonstrate how applying different types of spatiotemporal perturbations of specific intracellular signaling molecule(s) to an in-vitro tissue can uncover both intracellular and intercellular regulation as well as emergent phenotypes. Specifically, by controlling the levels of cAMP in chosen cells across an epithelial monolayer, either all cells or a small, localized group, we tease apart how the intracellular and intercellular regulation of cAMP

available at GitHub: https://github.com/mikechevy/emitter-receiver-paper.

**Funding:** This work was supported by the National Science Foundation, Award No: 1715108 to M.C. (Co-PI on the grant). The content and information does not necessarily reflect the position or the policy of the government, and no official endorsement should be inferred. A.H.N. was supported by the Department of Defense (DoD) through the National Defense Science & Engineering Graduate Fellowship (NDSEG) Program. The funders had no role in study design, data collection and analysis, decision to publish, or preparation of the manuscript.

**Competing interests:** The authors have declared that no competing interests exist.

impacts ERK signaling across cells in the tissue. Our approach could be used to contrast differences in signaling behavior between healthy and disease models of a given tissue type.

## Introduction

To maintain its ability to function physiologically, cells within tissues must coordinate through intimate collaboration and sharing of information to effectively respond to various perturbations so that the tissue may return to a homeostatic regime and function properly. These perturbations may include environmental stresses, nutrient availability, pathogens, genetic perturbations such as cancer [1], and physical perturbations such as wounds [2, 3] or changes in cell density [4]. While some perturbations may be approximately global across the tissue such as an in-flux of nutrients, others maybe more spatially localized such as a newly mutated cancerous cell or a wound.

Epithelial monolayers are an excellent model system for studying spatio-temporal tissue dynamics, both for applying perturbations and imaging its response [3, 4]. It is also an excellent initial model for studying diseased states like cancers in epithelial tissues [1]. In vivo, epithelial tissues line all cavities and organs of the human body, forming the surfaces of the eyes, the surfaces of the hollow tubes and sacs that make up the digestive, respiratory, reproductive, and urinary tracts, and the secretory cells and ducts of various glands. In addition to acting as physical and mechanical barriers to attack, epithelial cells also secrete many molecules with "cleansing" effects (lysozyme in tears, respiratory secretions, saliva, chemotactic and anti-microbial agents) [5]. And other crucial functions include integrity and control of permeability and epithelium polarity [6]. Fundamentally, the capability to perform these biochemical functions relies on coordination and communication among individual cells to produce coherent tissue output. To help achieve this epithelial cells share chemical signals such as cAMP, Ca2+, IP3, ATP and others through gap junctions [7] as well as secretion of molecules such as EGF through paracrine signalling [4, 8]. Furthermore, cell-cell communication is itself modulated by extracellular signals and intrinsic signals such as mitosis and apoptosis [9].

In addition to cell-cell communication, at the intracellular level coordination of multiple pathways within a cell are required to dictate context dependent tissue response and output. For instance, the cAMP/PKA and ERK pathways cross-talk within one epithelial cell and their coordinated function underlies an important swath of biology in epithelial cells including proliferation, apoptosis and growth [10, 11]. The second messenger cyclic-AMP (cAMP) is a ubiquitous signaling molecule whose synthesis by adenylyl-cyclase and degradation by phosphodiesterases occurs in all branches of life. In epithelial cells, cAMP signals are decoded primarily by 3 effectors: exchange protein directly activated by cAMP (Epac), cyclic-nucleotide gated channels including several $Ca^{2+}$ channels and pumps [12], and most prominently Protein Kinase A (PKA) [13]. Binding of cAMP to the regulatory subunit of PKA frees its catalytic units to phosphorylate hundreds of targets, regulating a vast swath of cellular physiology [14, 15]. Activation of the cAMP/PKA pathways inhibits Mitogen Activated Protein Kinase (MAPK) activation by growth factors in epithelial cells [16]. For instance, cAMP, through Epac and PKA, inhibits ERK [17, 18].

The dynamic intercellular and intracellular biochemical collaboration between the cAMP/PKA and ERK pathways has been relatively understudied even at a qualitative level in epithelial cells. cAMP can certainly cross gap junctions between neighboring cells and promote changes in cell fate [19–21]. However, we still have little understanding of how fast, how far and for

how long cAMP signals and subsequent downstream signals are shared across an epithelial tissue. Nor do we know how the time dependent signals vary from cell to cell. Therefore, studies that take into account this connectivity in one cell and explore the quantitative hallmarks and repercussions of their cell-cell communication under different contexts and conditions are needed to fully understand properties of epithelial tissues. The lack of progress to this question is mainly due to technological barriers where the ability to probe the dynamics of multiple pertinent pathways of single cells within a tissue in response to precise spatio-temporal inputs has been impossible. Recently, however, the tools for precise time-dependent perturbations of specific signaling molecules within single cells has undergone large leaps [22–27] allowing for new non-canonical perturbation that may enable a much greater understanding of the tissue as a whole [28, 29]. The ever growing list includes controllable transcription factors, enzymes, and membrane bound receptors [30, 31] to name a few. From a biological network perspective, the ability to perturb the level of signaling molecules from a given pathway and localized group of cells, allows one (with the proper reporters) to observe the signaling response both within the perturbed cells as well as their neighbors and any resulting phenotype. This can be a powerful tool for learning how the cells process the initial signal perturbation through signal processing, transduction and cell-cell communication. And crucially, recent progress has been made in quantitative cellular pathway reporters (such as Kinase Translocation Reporter (KTR) technology [32]) which can be used to measure the repercussions in individual cells across the tissue due to a given spatiotemporal perturbation. Overall, the combination of these new tools enable the collection of quantitative time-resolved data in single cells for basal tissue operation and also under rich stimulatory regimes. Such data are crucial for building and testing meaningful quantitative models [26] of tissues.

In this paper, we interrogate the effects of localized perturbations of cAMP on the cAMP/PKA and ERK pathways and cell-cell coupling. To achieve this, we develop a basic experimental approach to perturb cAMP levels in specific cells within an epithelial monolayer. Recent work [33, 34] have also applied cAMP perturbations to specific groups of cells within a monolayer, mainly a few localized cells, but their main focus was to prove that cell-cell coupling of PKA signaling occurs in the nearby neighboring cells. For our work we apply different types of spatiotemporal perturbations, including spatially uniform and spatially localized, to dissect the intracellular effects from those due to cell-cell coupling that impact signaling behavior within cells. Combined with pharmacological inputs, genetic perturbations, signal analysis, and computational modeling, we are able to infer how cell-cell coupling sculpts the intracellular cAMP and ERK signals. Our approach reveals how a dynamic cell-cell coupling state, via gap-junctions, impacts the intracellular cAMP/PKA and ERK pathways over time in locally perturbed cells.

## Results

To interrogate the effects of localized perturbations of cAMP on the cAMP/PKA and ERK pathways and cell-cell coupling, we developed a basic experimental approach, depicted in Fig 1. We employ Madin-Darby Canine Kidney (MDCK) epithelial cells for our studies, excellent for generating in-vitro epithelial monolayers [35]. To perturb cAMP levels, we apply bPAC, a blue light responsive adenylyl cyclase (Fig 1, top of left panel). bPAC is a naturally occurring bacterial molecule, consisting of a cyclase domain necessary for cAMP production that is preceded by a BLUF domain (sensors of blue-light using flavin adenine dinucleotide (FAD)) [22]. Stimulation of bPAC with blue light (<500nm wavelength) activates its cyclase activity rapidly (seconds). bPAC shutoff is also very rapid (< 30 s). The ability to stimulate a particular signaling molecule is powerful in that we know exactly how the system is getting

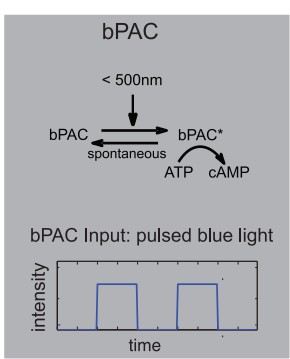
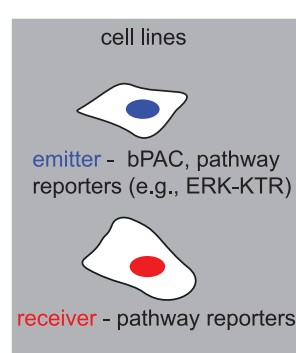
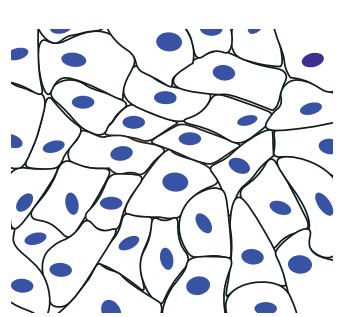
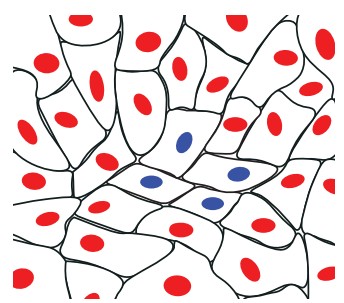

**Fig 1. Application of optogenetics to study the effects of localized cAMP perturbations in epithelial monolayers.** Optogenetic approach to study the effects of localized cAMP perturbations in monolayers: First panel: Top: bPAC, a blue light responsive adenylyl cyclase from bacteria. Bottom: The bPAC input is a sequence of spatially-uniform, constant amplitude (between 0 and 1) pulses of blue light. Second panel: Cell lines: The **emitter** has bPAC and pathway reporters while the **receiver** has only pathway reporters. Third panel: Monolayer of just emitters (spatially uniform cAMP perturbation). Fourth panel: Monolayer with a small emitter cluster surrounded by receivers (localized cAMP perturbation). We define a small emitter cluster as a concentrated group of 15 or less emitters.

perturbed internally, thus, the observed response is due to that singular perturbation. To perturb cAMP locally within a monolayer and measure its effects, we developed two cell lines in the MDCK cells, one with bPAC, we denote as **emitters**, and one without bPAC which we denote as **receivers** (Fig 1, second panel from left). Both cell lines have the same pathway reporters, e.g. ERK, PKA. To understand the effects of localized perturbations in the monolayer, it is informative to understand how the monolayer responds to a uniform perturbation, where every cell receives the same perturbation, approximately. This is achieved through pulsing spatially-uniform blue light input at a given amplitude (Fig 1, see bPAC input plot in bottom of left panel) onto an all-emitter monolayer (Fig 1, second panel from right). The same blue light input can then be applied to a monolayer with a small emitter cluster surrounded by receivers (Fig 1, right panel), and whose monolayer response can then be contrasted with that of the all-emitter monolayer.

Given our experimental approach, we dissect the intracellular effects from those due to cell-cell coupling. This is achieved through applying the experiments described in Fig 1 combined with pharmacological inputs, genetic perturbations, signal analysis, and computational modeling to infer how cell-cell coupling sculpts the intracellular cAMP and ERK signals within cells.

### The combination of bPAC and the ERK-KTR reporter provides quantitative single cell ERK activity measurements that are highly sensitive to intracellular cAMP/PKA dynamics

We first wanted to test the ability of bPAC to elicit signaling in canonical cAMP pathways for which reliable single cell-reporters exist. To do so, we subjected an all-emitter monolayer (Fig 1) containing either PRKACA or ERKKTR reporters to pulses of blue light inducing bPAC in every cell. Blue light-activated bPAC was clearly able to activate PKA as measured by the PRKACA reporter [11] (Fig 2A, see Materials and methods for details). It was also able to reduce ERK pathway activity, as measured by increased Nuclear/Cytoplasmic ratio in the ERK-KTR reporter (Fig 2B). For both reporters, consecutive pulses of blue light (40 ON/40 OFF pulses) resulted in repeatable reporter responses. Furthermore, activation of bPAC increased the frequency of $Ca^{2+}$ spikes [36] as expected (S1(A) Fig, measured using the GcAMP5h reporter [37, 38], see Materials and methods for details).

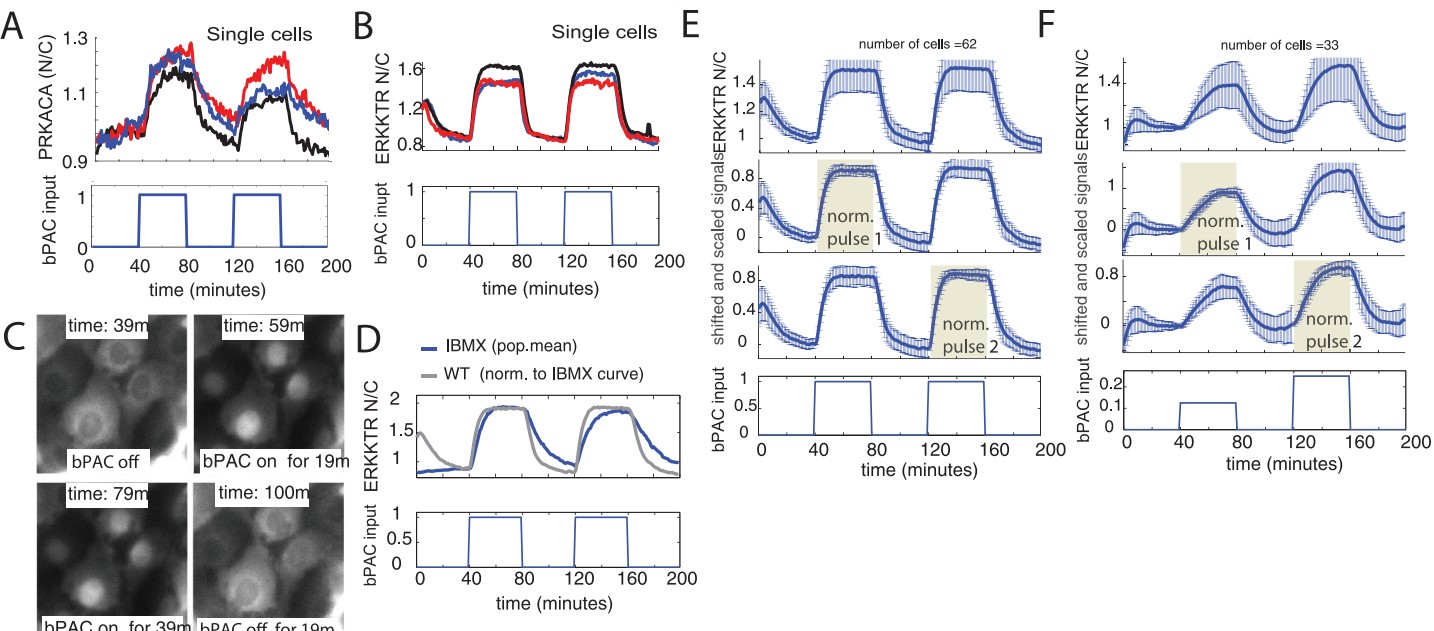

**Fig 2. bPAC induced emitters in all-emitter monolayers tend to have a simple, time-dependent ERK inhibition response.** (A) Periodic activation of bPAC (lower plot) results in periodic activation of PKA in single cells (upper plot). (B) Periodic activation of bPAC (lower plot) results in periodic ERK inhibition visualized by the increased ERK Kinase Translocation Reporter (ERK-KTR) nuclear to cytoplasmic ratio N/C in single cells (upper plot). (C) Time dependent snapshots of ERK-KTR reporter from experiment presented in (B). Here nuclear localization correlates with bPAC activity. (D) Periodic activation of bPAC (lower plot) in the presence of PDE inhibitor (IBMX) exhibits slower turn off of ERK inhibition (upper plot: blue, population mean) after each pulse than with no drug (upper plot, grey, normalized to blue for comparison). (E and F) Calculation of the population mean and standard deviation of the ERK-KTR N/C signal under different normalization approaches. Top 3 plots: For a given normalization approach, each single cell ERK-KTR N/C signal is shifted such that the signal at the beginning of the first pulse is zero and then normalized. Mean and standard deviation are calculated to just before the start of the second pulse. This process is the then repeated for the second pulse enforcing the standard deviation is zero at the beginning of each pulse. Top plot: mean ± standard deviation. Second plot from top: mean ± standard deviation calculated after each shifted signal is normalized by difference between the peak value of the signal during the first pulse and the value at the beginning of the first pulse. This removes the effects of peak amplitude variability in the first pulse. Third plot from top: same as prior normalization approach of the first pulse, but applied to the second pulse instead. This removes the effects of peak amplitude variability in the second pulse. Bottom plot: bPAC input pulse sequence. (E) All-emitter monolayer with identical bPAC input pulse amplitudes. (F) All-emitter monolayer with increasing bPAC input pulse amplitudes. For both (E) and (F), standard deviation is most constrained during the chosen pulse that removed the effects of peak amplitude variability (highlighted regions). Note that for some experiments one observes a decaying transient in the ERK-KTR N/C signal during the first 40 minutes when the the light is off (see single emitter examples in (B)). This occurs from exciting bPAC when searching for suitable cells to image with light that overlaps the spectrum that excites bPAC.

While all three reporters measured repercussions of the activation of bPAC by blue light, the ERK-KTR was the most robust and had the larger dynamic range. In addition, it also does not interact directly with cAMP and PKA thereby preventing potential altering of endogenous cAMP/PKA dynamics. Therefore, given that we are perturbing only cAMP in this work, we used the ERK-KTR as a downstream reporter of the cAMP/PKA pathway response to bPAC induction. Specifically, we quantified the ratio of ERK-KTR nuclear concentration to the ERK-KTR cytosolic concentration (ERK-KTR N/C), which is a measurement of ERK inhibition. An increase in cAMP levels increases PKA and Epac signaling, resulting in increased ERK inhibition and diminishing ERK's ability to keep ERK-KTR out of the nucleus [32]. As a result, the ERK-KTR nuclear concentration increases. The opposite occurs after bPAC inactivates and cAMP levels decrease (Fig 2C). Thus, the ERK-KTR N/C signal faithfully responds to bPAC activity and hence cAMP concentrations (Fig 2B).

Inhibition of phosphodiesterase (PDE) activity by IBMX leads to slower ERK-KTR N/C turn-off dynamics than cells without IBMX (Fig 2D). Therefore, the ERK-KTR N/C signal is able to mirror the slower degradation rate of cAMP ensuing from PDE inhibition [39], further

evidence that this signal is sensitive and responsive to the time-dependent cAMP concentration. Different blue light input pulses further corroborate the sensitivity of the ERK-KTR N/C signal to bPAC/cAMP dynamics for the operational regime needed for our investigations (S1 (B) Fig with discussion in S1 Text).

## bPAC induced emitters in all-emitter monolayers have a simple, first-order time-dependent ERK inhibition response

We sought to characterize the mean and variability across the population of single cell time-dependent ERK-KTR signals in all-emitter monolayer tissues for different pulsed experiments. For all experiments, we again use a sequence of two bPAC input pulses (40 min ON/OFF). For pulses of the same amplitude, the time-dependent mean of the ERK-KTR N/C response is very similar across pulses (Fig 2E and S2(A) Fig, top plot). For successive pulses of increasing amplitude, the maximum mean ERK-KTR N/C response tracks linearly (approximately) in the pulse amplitude (Fig 2F and S2(B) Fig, top plot). In all cases, significant variability across cells exists however. Although this variability is similar to what is observed from other inhibitions on the ERK pathway [32], we sought to better understand its origins.

For the initial mean and variability (standard deviation) calculations above, we calculated the standard deviation during each bPAC input pulse separately (including during shutoff). To ensure that we are comparing only variability that is driven by bPAC over each pulse and during shutoff, we first removed baseline variability at the beginning of a given bPAC input pulse. This is achieved by shifting each single-cell ERK-KTR N/C signal so that its value at the beginning of the pulse is zero (further details in Fig 2 and S2 Text).

Next we calculated the mean and standard deviation after normalizing each single cell signal by the peak response during the first pulse (further details in Fig 2 and S2 Text). We observe that that the variability about the mean during the first pulse becomes highly constrained (Fig 2E and 2F and S2(A) and S2(B) Fig, second plot from top (highlighted region)), significantly more so than during the second pulse. Similarly, applying the same normalization approach to the second pulse instead, we observe that the variability about the mean during the second pulse becomes highly constrained, significantly more so than during the first pulse (Fig 2E and 2F and S2(A) and S2(B) Fig, third plot from top (highlighted region)). This implies that the un-normalized variability (Fig 2E and 2F and S2(A) and S2(B) Fig, top plot) is dominated by variability in the peak amplitude of the ERK-KTR N/C signal (see S2 Text for detailed analysis and discussion). And aside from peak-amplitude variability, the single cell ERK-KTR N/C signals during a pulse tend to all have a very similar time-dependent behavior.

Overall, we observe that in the uniform all-emitter monolayer, the time-dependent ERK-KTR N/C response follows a simple over-damped rise upon bPAC activation at any amplitude we tested (see S1 and S2 Movies for visual examples). This characteristic response is general across cells in the all-emitter monolayer. Crucially, these results give us a reference for comparison when assessing signal behaviors in localized groups of emitter cells surrounded by receiver cells (Fig 1, small-emitter-cluster monolayer).

## Emitters in small emitter clusters can have a qualitatively different ERK inhibition signal during bPAC induction than those in an all-emitter monolayer

We next wanted to induce bPAC in small clusters of emitters surrounded by receivers to see how the emitter ERK-KTR N/C signal dynamics compared with the simple dynamics observed from emitters in an all-emitter monolayer. We generated monolayers with small emitter clusters (each emitter contains bPAC and ERK-KTR constructs) surrounded by receivers (each

receiver contains an ERK-KTR construct). These mosaic monolayers were created by mixing the two strains with much smaller proportions of emitter cells. In Fig 1, we defined an ideal small emitter cluster to be a concentrated cluster of emitters surrounded by receivers. While most emitters will be touching other emitters, we also allow 'stray' emitters, separated from the main group of connected emitters by a receiver cell, to be part of the cluster.

Experiments containing small clusters of emitter cells surrounded by receiver cells can have portions of emitters exhibit a distinct, dynamic overshoot in their individual ERK-KTR N/C responses. Fig 3A presents a striking example of the overshoot exhibited by many of the single emitters. Specifically, after the start of each bPAC input pulse, we observed a rapid increase in the inhibition of ERK activity in the emitter cells, where activity tended to partially recover over the following few minutes despite sustained blue light illumination. (Fig 3A, single emitters (bottom panel), emitter average (top panel, top plot)). For this example, overshoot of ERK inhibition occurs in both pulses of the bPAC input (40 min. pulse width, max amplitude). Furthermore, this emitter behavior in the small emitter cluster diverges from the simple first order response observed in emitters from all-emitter monolayer experiments for the same input pulse sequence (Fig 2E, for example).

While the small-emitter-cluster example in Fig 3A shows a majority of emitters exhibiting strong, dynamic overshoot in both pulses, other small emitter clusters may have smaller proportions of emitters respond with this type of overshoot (S3(A) Fig, left and center panels). Furthermore, some emitters may exhibit strong overshoot during one pulse, but exhibit a more flat response in the other. And some emitters may exhibit a flat response during both pulses (S3(A) Fig, left and center panels). That is, for a given small-emitter-cluster experiment, there can be heterogeneity in the ERK-KTR N/C signal across emitters, as well as across bPAC input pulses for a given emitter. It is both an emitter's ability to exhibit overshoot as well as the observed response heterogeneity within a given small emitter cluster that makes this overall phenotype intriguing.

For the example in Fig 3A, the ERK-KTR N/C overshoot within single emitters from the small emitter cluster tends to take anywhere from 8–15 minutes to complete, i.e. there is variability (Fig 3A, single emitter responses. Also see S3 Movie, includes description). On the other hand, all-emitter experiments typically show very little ERK-KTR N/C overshoot, statistically (Fig 2E and 2F, with single emitter plots in Fig 2B). When they do, it tends to be at a much slower timescale and a more subtle, slow overshoot (S1(C) Fig, fourth single cell plot, for example) as compared to the dramatic, dynamic overshoot in many of the emitters in the small-emitter-cluster examples.

## Statistical analysis of emitter ERK-KTR signals supports the qualitative differences observed between all-emitter and small-emitter-cluster experiments

Next we wanted to statistically quantify the qualitative differences observed in the ERK-KTR N/C signal between emitters in all-emitter experiments and those in small-emitter-cluster experiments. The first statistical metric we calculated for each emitter signal was the time to reach peak ERK-KTR N/C signal amplitude during the bPAC input pulse, denoted as *time-to-max* (see Fig 3C for illustration). This is a proxy for when overshoot occurs in emitters or when it reaches steady state when there is no overshoot. The *time-to-max* tends to be significantly slower on average for emitters in an all-emitter monolayer than for emitters in a small emitter cluster surrounded by receivers (Fig 3D, left panel, mean error bars in green represent 95 percent confidence interval of the mean, compare means across groups). Statistically, the all-emitter experiment with the lowest mean (bPAC input pulse amplitudes of one (max)) has

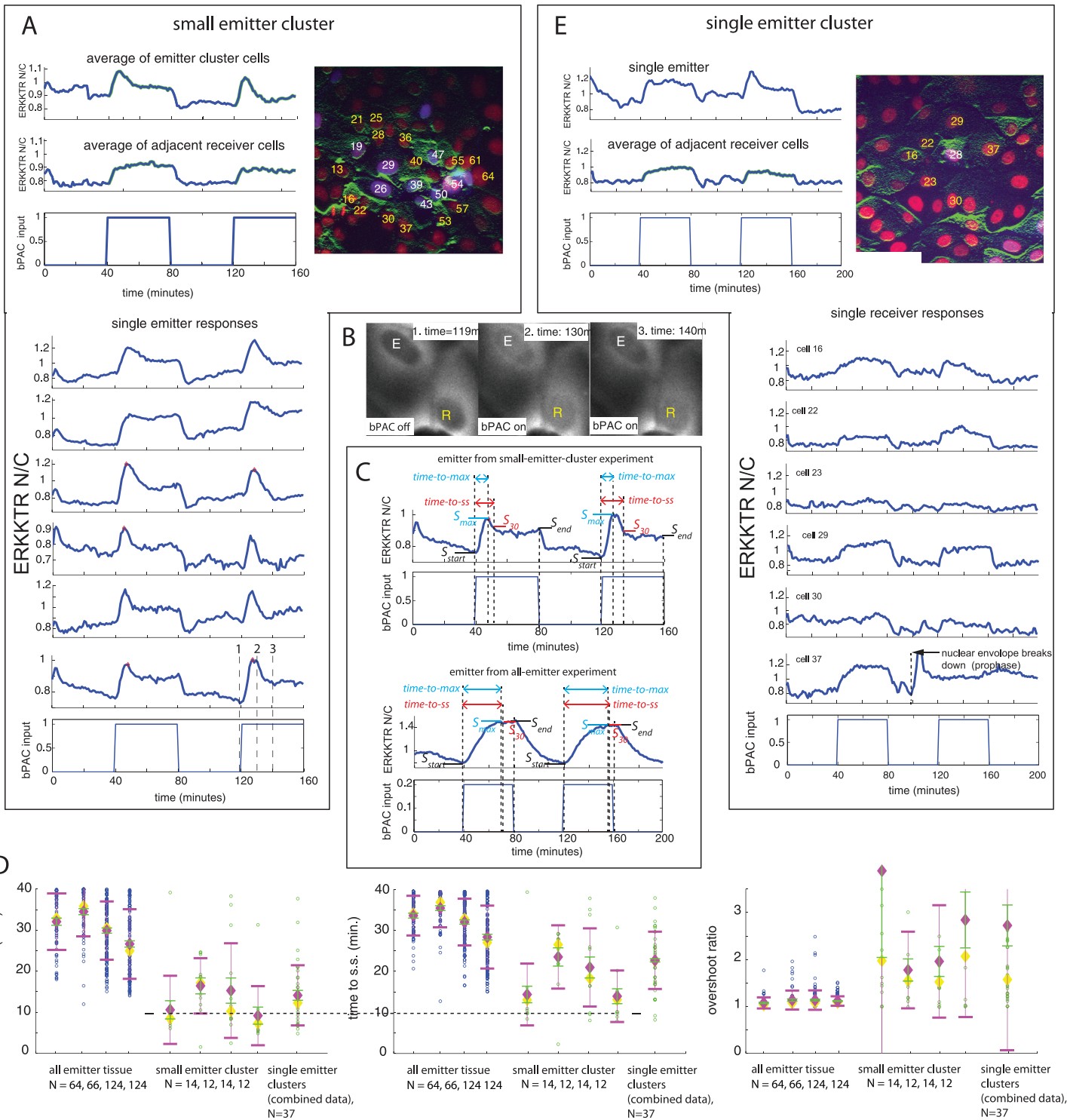

**Fig 3. Single cell emitter ERK-KTR N/C signals from small-emitter-cluster monolayers can be qualitatively different from emitters in all-emitter monolayers.** Images of monolayers in (A) and (E): merged image of green (filtered ERK-KTR with horizontal line detector filter), red (nuclear maker), blue (nuclear bPAC marker) where emitter nuclei are purple (red+blue), and receiver nuclei are red. Labeled nuclei: white label (emitter), yellow label (receiver). (A) Results for a small-emitter-cluster monolayer. Left top panel: top plot: small-emitter-cluster average of ERK-KTR N/C signal, middle plot: adjacent receivers average of ERK-KTR N/C signal, bottom: bPAC input. Right image: small emitter cluster. Bottom panel: single cell emitter measurements of ERK-KTR N/C signal from the small-emitter-cluster experiment. (B) Time dependent snapshots of ERK-KTR reporter from experiment presented in (A) for the emitter (white 'E' label) whose response is plotted in the bottom single cell emitter plot in (A) (dashed vertical lines depict snapshot times) and an adjacent receiver (yellow 'R' label). (C) Illustrations depicting measurements used for time-to-max, time-to-ss, and overshoot-ratio statistical metrics. The time-to-max is the time to reach $S_{max}$, max signal during pulse. The time-to-ss is the time to reach $S_{30}$, i.e. when the signal is first within 30 percent of $S_{end}$ (end-of-pulse signal value) relative to $S_{max}$. Given $S_{start}$ (start-of-pulse signal value), the overshoot ratio is $\frac{S_{max}-S_{start}}{S_{end}-S_{start}}$. Top

panel: Measurements of the ERK-KTR N/C signal for a single emitter from small-emitter-cluster experiments (including that in (A)). Bottom panel: Measurements of the ERK-KTR N/C signal for a single emitter from all-emitter experiments (including those in Fig 2E and 2F). As shown in both panels, measurements are taken during each bPAC input pulse. (D) Scatter plots of time-to-max times, time-to-ss times, and overshoot-ratio for single cell ERK-KTR N/C signals taken from all-emitter experiments (blue, with the two rightmost ones from experiments presented in Fig 2E and 2F), small-emitter-cluster experiments (green, with the leftmost one from experiment presented in (A)), and combined data for 10 single-emitter-cluster experiments (some experiments have more than one single emitter cluster in the field of view). Mean (magenta diamond) ± standard deviation (magenta error bars), error bars in the mean (green, standard deviation/$\sqrt{N}$) representing 95 percent confidence interval in the mean, and median (yellow diamond) are also plotted. Number of data points $N$ used are displayed below each group label. (E) Results for a single-emitter-cluster experiment. Left panel: top plot: single emitter ERK-KTR N/C signal, middle plot: adjacent receivers average of ERK-KTR N/C signal, bottom plot: bPAC input. Right image: single emitter cluster. Bottom panel: plots of ERK-KTR N/C signal for adjacent receivers, and bPAC input (bottom plot).

a greater mean than the pooled small emitter cluster data ($p < 10^{-5}$ (see Materials and methods for details)).

Next, for each emitter, we calculated the approximate steady-state convergence time after reaching the maximum signal amplitude. We defined this as the time the signal drops to within 30 percent of the end of pulse signal amplitude relative to the max signal amplitude, and denoted this time as *time-to-ss* (see Fig 3C for illustration). Thus, for a given emitter, *time-to-ss* ≥ *time-to-max*. For the all emitter experiments, we see very little change in the mean between the *time-to-ss* and the *time-to-max*, since the mean *time-to-max* is already high and cannot exceed 40 minutes (Fig 3D, middle panel, all-emitter data). This is also because for many of these emitters the *time-to-max* represents reaching steady-state not overshoot. For the small-emitter-cluster experiments we see the mean *time-to-ss* shift by less than 10 minutes on average (across experiments) relative to the *time-to-max* (Fig 3D, middle panel, small-emitter-cluster data, dashed line across left and middle panels serves as a reference). Because many of the emitters exhibit overshoot, this give an average sense of the overshoot dynamics, with plenty of variability about this mean (see S3(C) Fig for *time-to-ss−time-to-max* statistics, mean and variability).

For a final metric we wanted to quantify the magnitude of the overshoot across experiments. For this, we applied an *overshoot ratio* metric defined as $\frac{S_{max}-S_{start}}{S_{end}-S_{start}}$ where $S_{max}$ is the max signal value during the bPAC pulse, $S_{start}$ is signal value at the start of the pulse, and $S_{end}$ is the signal value at the end of the pulse (see Fig 3C for illustration). For all-emitter experiments the majority of the overshoot-ratio measurements are near one, with a few outliers (Fig 3D, right panel, all-emitter data). While for emitters in small emitter clusters, there is a wide range of overshoot-ratio measurements ranging from near one, similar to the all-emitter experiments, to very large overshoot-ratio (outside the range plotted) (Fig 3D, right panel, small-emitter-cluster data, some data points are outside the range plotted). Statistically, the pooled small emitter cluster data has a greater mean for the *overshoot-ratio* than any all-emitter experiments ($p < .012$ (see Materials and methods for details)).

In summary, emitters in small-emitter-cluster experiments have a larger overshoot ratio on average than those in all-emitter experiments. And for the emitters in small-emitter-cluster experiments, the overshoot dynamics on average converge within 10 minutes after the peak signal is reached with some emitters having faster dynamics than the average and some having slower dynamics.

## Isolated single emitters also can exhibit dynamic overshoot in their ERK inhibition signal during bPAC induction

Importantly, for the small-emitter-cluster experiment, the surrounding receiver cells exhibit a response in their ERK-KTR N/C signal indicating coupling between emitter and adjacent receiver cells, all of which touch emitters. And on average, the receiver cells do not exhibit overshoot (Fig 3A, top panel, middle plot). An example of overshoot in the ERK-KTR nuclear

concentration is captured for a single emitter along with its coupling to an adjacent receiver which exhibits no overshoot (Fig 3B). This motivated us to induce bPAC in the most minimal small-emitter-cluster, a single emitter, i.e. single emitter cluster, where there are no direct physical interactions or contact with other emitters. We conjectured that it might also exhibit overshoot. That is, the overshoot in emitters is somehow fundamentally related to the coupling of the ERK-KTR N/C signal to receiver cells, possibly through cAMP coupling. For this paper, we examine single emitters that are isolated by at least three or more receiver cells between itself and other single emitters or small emitter clusters (Fig 3E, image: single emitter (white label), adjacent receiver cells (yellow labels)). Indeed, when we apply a two pulse bPAC input, both at max amplitude, we observe overshoot in the ERK-KTR N/C signal, most prominently during the second pulse (Fig 3E, single emitter plot. Also see S4 Movie, includes description). The first pulse yields a quick minimal overshoot, one could argue, while the second pulse yields a slower more pronounced overshoot, similar to those observed in some of the single cells in the small cluster emitter experiment (Fig 3A). While the emitter does exhibit overshoot, the response is quite heterogenous across pulses. As with small emitter clusters, there are also single emitter clusters that exhibit overshoot across pulses, as well as ones that exhibit no overshoot (S3(A) Fig, right panel, and see S3(B) Fig for images of time snapshots of emitter ERK-KTR response for emitters that exhibit overshoot).

Similar to the small-emitter-cluster experiments, pooled data from numerous single-emitter-cluster experiments show that *time-to-max* in the ERK-KTR N/C signal during a bPAC input pulse for single emitters tends to be lower, on average, than emitters in the all-emitter experiments (Fig 3D, left panel, single-emitter-cluster data, mean error bars in green represent 95 percent confidence interval of the mean, compare means across groups. Statistically, the all-emitter experiment with the lowest mean (bPAC input pulse amplitudes of 1 (max)) has a greater mean than the pooled single emitter cluster data ($p < 10^{-5}$ (see Materials and methods for details)). In addition, both *time-to-ss* and *overshoot ratio* for the pooled single-emitter-cluster experiments behaved similarly to the small emitter cluster data (compare in Fig 3D, middle panel (*time-to-ss*) and right panel (*overshoot ratio*), and in S3(C) Fig for *time-to-ss–time-to-max* statistics, mean and variability). Statistically, the pooled single emitter cluster data has a greater mean for the *overshoot ratio* than any all-emitter experiments ($p < .012$ (see Materials and methods for details)).

For single-emitter-cluster experiments, unlike the emitter, the average ERK-KTR N/C signal of the adjacent receiver cells tend to not exhibit a pronounced overshoot during either pulse (Fig 3E, top panel, middle plot and S3(B) Fig, plots of average of adjacent receivers). This also tends to be the case for individual receivers (Fig 3E, lower panel: single receiver cells). For the single emitter example in Fig 3E, we also observe a rare example of an adjacent receiver (cell 37) entering prophase of mitosis losing its nuclear envelope and, thus, the ERK-KTR is cytosolic and unresponsive. Later in this work we will discuss how cells in this state receive little to no coupled ERK inhibition signal from the emitter.

Finally, we wanted to look at the at ERK-KTR N/C signal propagation from single emitter clusters as a function of steady-state amplitude taken at the end of a 40 minute pulse. We found that steady-state receiver signals follow a decreasing trend with increasing distance from the emitter (S3(D) Fig).

## Coupling of ERK-KTR N/C signal requires cAMP transport through gap junctions controlled by PKA

We next wanted to explain the overshoot observed in the emitter ERK-KTR N/C signal for small emitter and single emitter clusters as well as the behavior of the ERK-KTR N/C signal

propagation to receiver cells. We hypothesized that the cAMP produced by bPAC was diffusing through gap-junctions to the surrounding receiver cells which were acting as sinks, depressing maximal cAMP concentration in the emitter cells. Indeed, inhibiting gap-junctions inhibits the receiver ERK-KTR N/C signal (Fig 4A (average signals) and S4(B) Fig (mean with error bars)), implicating gap junctions as the main coupling mechanism, and thus cAMP as the coupled signaling molecule between emitters and receivers. Interestingly, the average emitter ERK-KTR N/C signal decay after the bPAC input pulse shuts off (Fig 4A) is very similar to all-emitter experiments with IBMX (compare Fig 2D with S4(A) Fig). This is likely because carbenoxolone is known to inhibit PDEs resulting in slower cAMP decay.

Unexpectedly, when we add PKA inhibitor (H89, 10 $\mu$M), the cAMP coupling from emitters to receivers is greatly reduced (Fig 4B (average signals) and S4(C) Fig (mean with error bars)), suggesting gap junction activity is related to PKA. Indeed, recent work [40, 41] found that PKA phosphorylates gap junctions, as an Ezrin/PKA/gap-junction (connexin 43) complex, through cAMP activation, thereby controlling gap-junction activity (permeability). Thus, PKA inhibitors are inhibitors of gap junction activity. In addition, we also observe that the average emitter ERK-KTR N/C signal did not exhibit fast, dynamic overshoot (Fig 4B), thus, strengthening support for our hypothesis that gap-junctions are responsible for the strong, fast overshoot observed in many emitters from small emitter clusters.

While both PKA and Epac inhibit ERK [17, 18] and thus both contribute to the ERK-KTR N/C signal during cAMP induction through bPAC (see S3 Text and S5(A) and S5(B) Fig), Epac is not known to inhibit gap-junctions. Consider a small-emitter-cluster, two-pulse bPAC-input experiment where an inhibitor is added halfway between the two pulses. Our results from Fig 4B, predict that during the second pulse of a PKA inhibition experiment of this type, the emitters will show little to no dynamic overshoot in the ERK-KTR N/C signal as well as negligible coupling to receivers, which is what we observe (Fig 4C). However, for an Epac inhibition experiment the emitter should still be capable of exhibiting dynamic overshoot as well as coupling to receivers, which is what we observe (Fig 4D), providing further evidence of gap junctions being the culprit causing the observed overshoot.

As a final control, these cell-cell coupling insights make the prediction that small emitter clusters and single emitter clusters that are surrounded by receivers that do not form gap junctions (MDCKII cell line) should exhibit emitter ERK-KTR N/C signals that resemble all-emitter monolayers. Indeed that is what occurs in both small emitter clusters (Fig 4E and S4(D) Fig) and single emitter clusters (S4(E) Fig). As expected, the *time-to-max* statistics also shift upwards to the all-emitter regime (Fig 4F, left panel, small-emitter-cluster and single-emitter-cluster data, mean error bars in green). Statistically, the pooled data of the small emitter clusters surrounded by receivers with no gap junctions has a greater mean than the pooled small-emitter-cluster data from Fig 3D (left panel) ($p < 10^{-5}$ (see Materials and methods for details)). In addition, as might be expected both *time-to-ss* and *overshoot ratio* for the small emitter clusters and single emitter clusters surrounded by receivers that do not form gap junctions behave similarly to the all-emitter data (Fig 4F, middle panel (*time-to-ss*) and right panel (*overshoot ratio*), small-emitter-cluster and single-emitter-cluster data). Statistically, pooled small-emitter-cluster data from Fig 3D (right panel) has a greater mean for the *overshoot-ratio* than the data for the pooled small emitter clusters with receivers that have no gap junctions ($p < .012$ (see Materials and methods for details)).

Altogether, these experimental results establish that the coupling of cAMP from emitters to receivers through gap-junctions is what gives rise to the overshoot dynamics observed in many of the emitters in the small-emitter-cluster and single-emitter-cluster experiments.

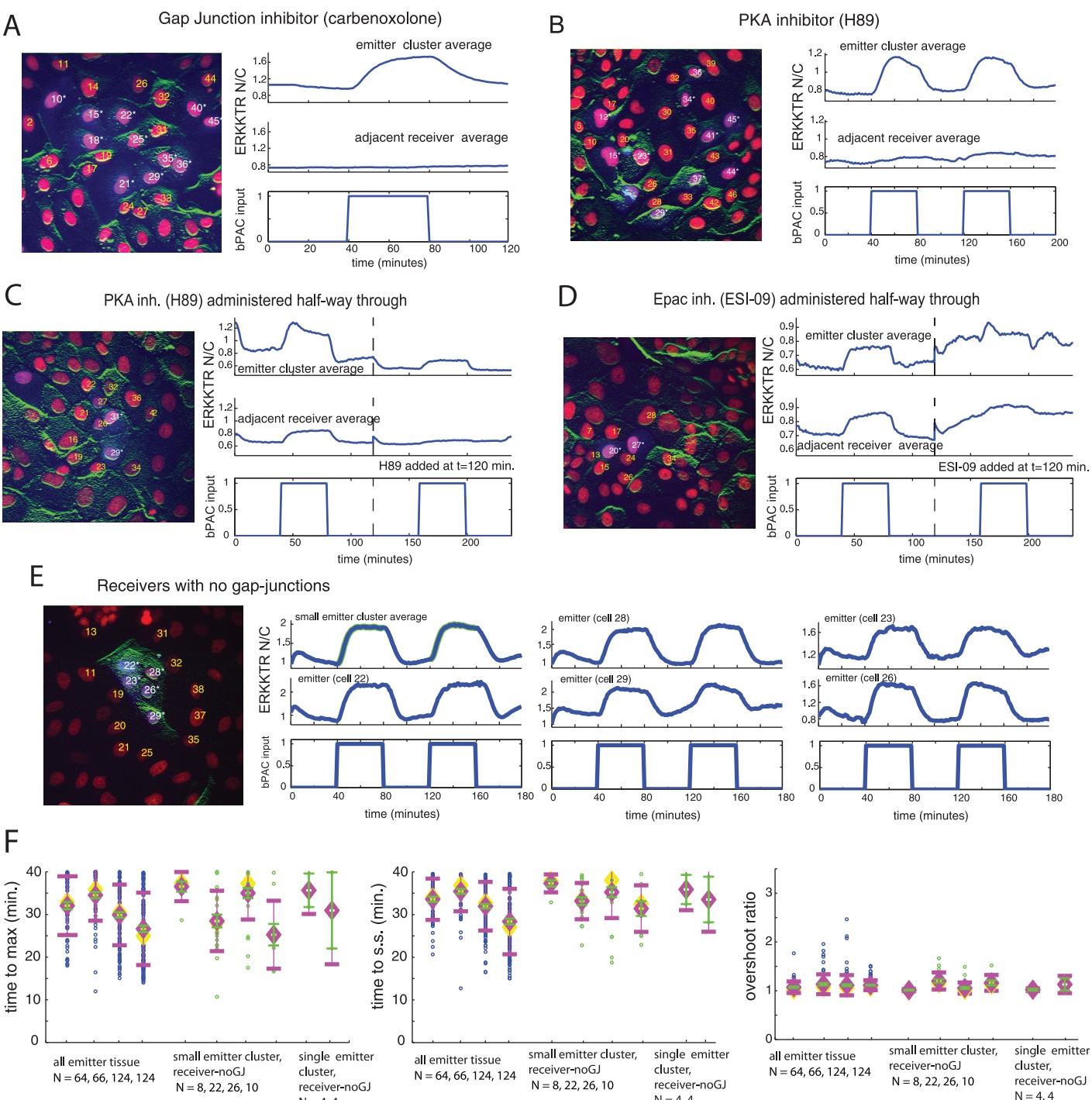

**Fig 4. PKA and cell-cell coupling of cAMP through gap junctions are required for the dynamic overshoot observed in the ERK-KTR N/C signal of bPAC induced emitters from small emitter clusters. Single cell ERK-KTR N/C signals in all-emitter monolayers are representative of the dynamics due to intracellular regulation**. (A-D) Left image depicts emitter cluster (emitter cells—purple nuclei with white labels, adjacent receiver cells—red nuclei with yellow labels). Right Panel: top plot: average small-emitter-cluster ERK-KTR N/C signal, middle plot: average adjacent receivers ERK-KTR N/C signal, bottom plot: bPAC input pulse sequence (max amplitude). (A) Small emitter cluster experiment with gap-junction inhibition (carbenoxolone): Emitter-receiver coupling is lost. (B) Small emitter cluster experiment with PKA inhibition (H89, 10 $\mu$M): Emitter-receiver coupling is inhibited. (C and D) Testing how Epac and PKA inhibitors affect ERK under bPAC activation for small-emitter cluster experiments. Cells are imaged during a pulsed bPAC input sequence (40 minutes off, 40 minutes on, 40 minutes off). An inhibitor is then added and the bPAC input sequence is repeated while the same cells are imaged. (C) Small emitter cluster driven by a sequence of bPAC input pulses. PKA inhibitor (H89, 10 $\mu$M) is added half way through. (D) Small emitter cluster driven by a sequence of bPAC input pulses. Epac inhibitor (ESI-09) is added half way through. (E) Experimental results for small emitter cluster surrounded by receivers that do not form gap junctions (MDCKII cells). Left panel: Image of small emitter

cluster where receivers do not have ERK-KTR but have a nuclear marker (red nuclei, yellow label). Panels to the right of image: Average emitter ERK-KTR N/C signal and single cell ERK-KTR N/C signals for the small emitter cluster. (F) Scatter plots of time-to-max times, time-to-ss times, and overshoot ratio for single cell ERK-KTR N/C signals taken from the same all-emitter experiments in Fig 3D, small-emitter-cluster experiments (surrounded by receivers that do not form gap junctions), and single-emitter-cluster experiments (surrounded by receivers that do not form gap junctions). Mean (magenta diamond) ± standard deviation (magenta error bars), error bars in the mean (green, standard deviation/$\sqrt{N}$) represent 95 percent confidence interval in the mean, and median (yellow diamond) are also plotted. Number of data points $N$ used are displayed below each group label.

## A simple model of the all-emitter experiments

The ERK-KTR N/C signal overshoot dynamics that can be observed in emitters of small-emitter-cluster and single-emitter-cluster experiments, but not for emitters in all-emitter experiments, inspired us to develop a basic model that can account for these observations.

We first wanted to construct a simple intracellular model based on the experimental observations above. To summarize, the rise and decay dynamics of the emitter ERK-KTR N/C signal are very similar between all-emitter experiments (Fig 2E and 2F, for example) and small-emitter-cluster experiments surrounded by receivers with no gap junctions (Fig 4E and S4(D) and S4(E) Fig). They all follow simple first-order like dynamics. Furthermore, if, for example, we apply a pulse sequence of increasing amplitude to an all-emitter monolayer with PKA inhibition (H89) (S5(C) Fig), which has reduced cell-cell coupling, we observe a similar response in the ERK-KTR N/C signal as that for the a similar no-drug all-emitter experiment (Fig 2F). The *time-to-max* statistics are similar as well (S2(E) Fig). All these observations suggest that we are observing how the intracellular regulatory circuit sculpts the ERK-KTR N/C signal. And even though there is cell-cell coupling for the wild-type all-emitter experiments, there should be no strong net-flux, on average, between cells, unlike the emitter-receiver coupling in the wild-type small-emitter-cluster experiments. This is because all cells are producing cAMP in the all-emitter experiments.

In constructing a simple intracellular circuit, we consider known feedbacks on cAMP (see intracellular circuit in S5(D) Fig, a literature-based model with feedbacks on PDE from [42]), including PKA and Calmodulin Kinase II feedback. Because rise and decay dynamics of the ERK-KTR N/C signal are very similar for the PKA inhibited all-emitter experiments and the wild-type all-emitter experiments, this suggests that PKA might not have strong feedback through PDEs on cAMP. In addition, recent work [42] has uncovered a feedback through Epac to Calmodulin Kinase II to PDE4. The bPAC induced all-emitter experiment that inhibits Epac halfway through shows a similar time-dependent shape of the ERK-KTR N/C signal before and during Epac inhibition (S5(B) Fig). It is possible that enough Epac might be active during inhibition to still provide negative feedback through Calmodulin Kinase II onto to PDEs to ensure this. Or it is possible that the feedback through Calmodulin Kinase II is not significant in the regime we are testing the system.

Given that there is not much difference in the ERK-KTR N/C signal dynamics with or without the inhibitors (PKA or Epac), we will consider a more simple intracellular circuit where the cAMP degradation is driven by a simple rate constant, $\gamma_{pde}$. For this case, the cAMP degradation term is simply $[cAMP]\gamma_{pde}$, the concentration multiplied by the rate constant, i.e. a simple first order decay (Fig 5A, intracellular circuit, see S4 Text for further details of the computational model).

We model the ERK-KTR N/C signal as a two step process. First, the combined effects of cAMP driving both Epac and PKA to deactivate ERK (through Rap1 [10]) are represented by a simple cAMP-dependent deactivation rate of ERK. Second, the effect of ERK driving EKT-KTR into the cytosol is represented as simple ERK-dependent, cytosolic translocation

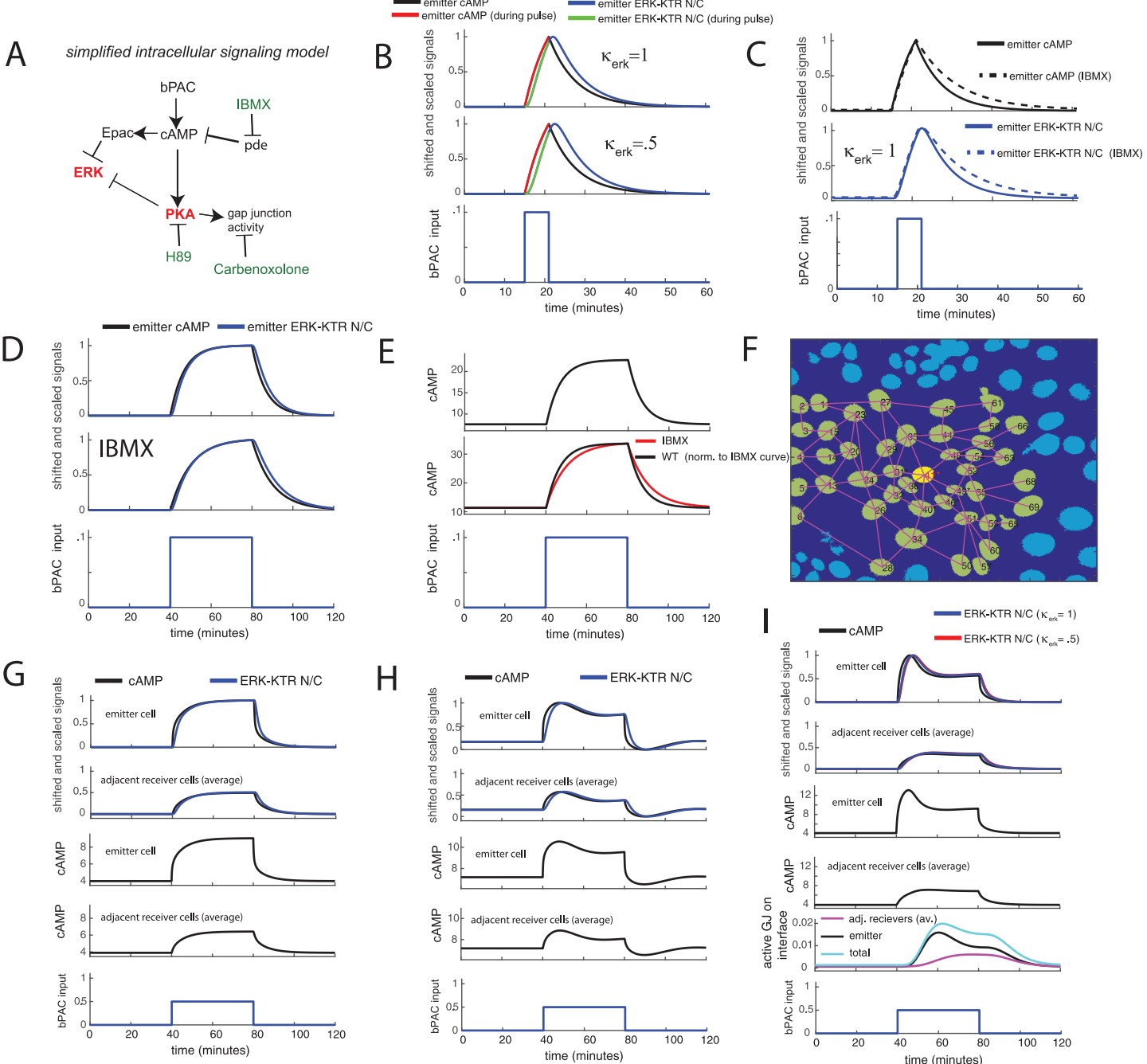

**Fig 5. Model requires gap junction regulation with a delay to recapitulate difference in ERK-KTR N/C signals between emitter and adjacent receiver cells.** (A) Illustration of simplified intracellular pathway. (B-E) Modeling results for simplified intracellular pathway with no feedbacks in an all-emitter monolayer. Model is fitted to ERK-KTR N/C data for 6 minute bPAC input pulses applied to an all-emitter monolayer (S1(B) Fig, IBMX, population average). ERK-KTR N/C and cAMP signals shifted to zero and scaled for comparison, with the peak of the emitter signal at 1 and minimum at zero. (B) ERK-KTR N/C signal and predicted cAMP signal driven by 6 minute bPAC input pulse. We present results for two values of the deactivation rate of ERK $\gamma_{erk}$ using the expression $\kappa_{erk}\gamma_{erk}$ where $\kappa_{erk}$ is a scaling factor and $\gamma_{erk}$ is a fitted deactivation rate. Top plot: fitted model ($\kappa_{erk} = 1$). Middle plot: same fitted model except for a slower ERK deactivation rate reduced by a factor of 2 ($\kappa_{erk} = .5$). (C) Fitted model results ($\kappa_{erk} = 1$) for ERK-KTR N/C signal and predicted cAMP signal, with and without IBMX. (D) Predicted 40 minute bPAC input pulse driven ERK-KTR N/C and cAMP signals (with and without IBMX). Signals shifted to zero and normalized for comparison. (E) Modeled cAMP signal (top plot), and with PDE inhibitor (IBMX, middle plot). Same signals as in (D) but not zero shifted and normalized. (F) Multicellular system to be modeled with 1 emitter (yellow nucleus, red label) and receiver cells (light green nuclei, black label). Lines connecting nuclei represent direct cell-cell communication through gap-junctions. (G-I) Single-emitter-cluster modeling using system in (F): For top two plots, the ERK-KTR N/C and cAMP signals are shifted to zero and scaled for comparison, with the peak of the emitter signal at 1 and the minimum at zero. The receiver signals are multiplied by the same scaling factor as the emitter to maintain the same relative amplitudes (proportionally). (G) Results for the simplified intracellular pathway with constant gap junctional permeability. (H) Results for the intracellular PKA feedback model

through PDE that produces overshoot (in all-emitter monolayer) with constant gap junctional permeability. (I) Results for the simplified intracellular pathway with cAMP/PKA regulation of gap junctions with a delay. Corresponding dynamics of the active gap junctions on the emitter/receiver interfaces (averaged) is also included with a breakdown of the cAMP/PKA driven contributions from the emitter and the average of the adjacent receivers.

rate of ERK-KTR (details in S4 Text). The model was fit to data from the 6 minute bPAC input pulse experiments discussed in S1 Text (see S6(A)–S6(C) Fig for model/fit). We present two fitted examples whose ERK deactivation rate differs by a factor of two (Fig 5B and 5C). Both cases show a slightly delayed ERK-KTR N/C response relative to the modeled cAMP signal, with the slowest ERK deactivation rate ($\kappa_{erk}$ = .5) case exhibiting the most delay. While a little sluggish relative to the cAMP signal, we will show that the fitted ERK-KTR N/C signals mirror the salient qualitative features of the cAMP signal such as overshoot. We also simulate the system in the presence of PDE inhibitor (IBMX) which expectedly decreased the decay rates (Fig 5C, IBMX cases) of the signals upon turnoff.

Results for this fitted simple intracellular model, including the model of the ERK-KTR N/C signal, are presented for an all-emitter monolayer for a 40 minute pulsed bPAC input which captures the basic first order rise and turn off of the cAMP signal and ERK-KTR N/C signal (Fig 5D and 5E and S6(D) and S6(E) Fig, with and without the PDE inhibitor IBMX) as well as PKA (S6(F) Fig). With this model of the intracellular circuit, we turn our focus to modeling the cell-cell coupling mechanism which we deduced to be the major factor for the strong overshoot of the ERK-KTR N/C signal observed in the small-emitter-cluster and single-emitter-cluster experiments.

## Determining a simple regulated gap-junction flux model for the cell-cell coupling of cAMP

The data measured from the experiments presented in Figs 2–4 implicates gap-junctions as the mechanism for the overshoot observed in the ERK-KTR N/C signal in single emitters from single-emitter-cluster experiments. However, our experiments do not measure the gap-junction activity at cell-cell interfaces that would drive this phenotype. Here we explore computational gap-junction models that could support our experimental observations. As a foundation, we apply a standard gap-junction flux model [43, 44] where the model for the flux $F_{c_{ij}}$ of cAMP through gap-junctions at the interface of cell $i$ and cell $j$ is

$$F_{c_{ij}} = \rho_{ij} \times ([cAMP^j] - [cAMP^i]) \tag{1}$$

Here $\rho_{ij}$ is the effective gap-junctional cAMP permeability (can be time-dependent) at the interface of cell $i$ and cell $j$. And $[cAMP^j] - [cAMP^i]$ is the difference of cAMP concentrations at the gap-junction interface and dictates the direction and magnitude of the flux.

We first applied a simple cell-cell coupling model of constant gap junctional permeability, i.e. the coupling capability is constant over time. For this case we set $\rho_{ij} = \omega_{ij}k_{gj}$, where $\omega_{ij} = 1$ when cells $i$ and $j$ interface, otherwise $\omega_{ij} = 0$. And $k_{gj}$ is a constant resulting in constant gap junctional permeability across all cell-cell interfaces. Thus for this case, $\rho_{ij}$ is not time-dependent. For a single emitter surrounded by receivers pictured in Fig 5F, we find that this model cannot reproduce the strong overshoot (Fig 5G and S6(G) Fig, modeled over a range of constant gap junctional permeability values to test different coupling strengths). Next we tested a strong intracellular PKA feedback through the PDEs that is capable of producing dynamic overshoot in an all-emitter monolayer (S6(I) Fig). While we already know it is not representative of the actual intracellular system, we wanted to see how the receiver cells responded. When we run the single-emitter-cluster case from Fig 5F, we get the expected overshoot in the

emitter, but the receiver mirrors the emitter with overshoot (Fig 5H and S6(H) Fig, modeled over a range of constant gap junctional permeability values to test different coupling strengths), not the simple response with no overshoot observed by the adjacent receivers in the single-emitter-cluster experiment in Fig 3E. Thus, this simple cell-cell coupling model of constant gap junctional permeability is insufficient.

From a regulation perspective, the simple gap junction model imposes a negative regulation on the single emitter intracellular cAMP signal since cAMP is leaving the emitter through gap junctions to the surrounding receivers. However, this is a time independent regulation. It is well known, especially in feedback systems, that delayed negative regulation on a signal is a mechanism that can cause overshoot in that signal [45]. With this in mind, we hypothesized the overshoot to be due to cAMP/PKA dependent gap-junction regulation with a delay. When cAMP levels go up there is an increase in gap-junction permeability with about a 10 minute delay to reach a maximum permeability, potentially related to either transport of connexin 43 onto the membrane and/or the formation of additional PKA/ezrin complexes tethered to connexin 43. At about 10 minutes this 'opens the flood gates' enabling greater flux between cells thus allowing for a strong overshoot in the emitter cells. A simple means of capturing this effect involves a series of finite delay steps initiated by cAMP/PKA to activate gap-junction molecules. Here, $GJ^i_{na}$ is the inactive gap junction molecule, in the $i$th cell of the monolayer, which goes through a series of $N$ intermediate delay steps where $GJ^i_{d_k}$ represents the gap junction molecule at the $k$th delay step, for $k = 1,2,..,N$. After the final delay step the molecule is then in the active state, $GJC^i_a$, a fully formed active gap junction. This is captured by the chemical equations

$$GJ^i_{na} \xrightarrow{f(PKA^i_{on})} GJ^i_{d_1} \xrightarrow{\frac{N}{\tau_{gj}}} GJ^i_{d_2} \xrightarrow{\frac{N}{\tau_{gj}}} \quad \cdots \cdots \quad \xrightarrow{\frac{N}{\tau_{gj}}} GJ^i_{d_N} \xrightarrow{\frac{N}{\tau_{gj}}} GJC^i_a \qquad (2)$$

where $\sum_{k=1}^{N} \frac{\tau_{gj}}{N} = \tau_{gj}$. Here $\tau_{gj}$ is the mean time it takes for a group of $K$ gap-junction molecules in the first delay state $GJ^i_{d_1}$ to reach the active state $GJC^i_a$. Here the mean delay $\tau_{gj}$ could represent transport and/or Ezrin/PKA/connexin complex formation prior to becoming active and coupling to a neighboring cell. For $N = 1$ steps, the time dependent accumulation of the group into the active state can be expressed as $K(1 - \exp(-t/\tau_{gj}))$, i.e. highly dispersed in activation times. For large N, it is a switch like (perfect delay) response, $Ku(t - \tau_{gj})$, a step function where every molecule becomes active at the same time (no dispersion in activation times). For both cases we are assuming the group starts in state $GJ^i_{d_1}$ at $t = 0$. Intermediate values of $N$ will have intermediate dispersion of activation times relative to the two extremes presented (see S6(J) Fig for examples). For those molecules in the active state, they can then de-activate (endocytosis, for example) back to the inactive state via the chemical equation

$$GJC^i_a \xrightarrow{\gamma_{gj}} GJ^i_{na} \qquad (3)$$

The delayed regulation model in cell $i$ and in cell $j$ modulates the effective gap junctional cAMP permeability, $\rho_{ij}$, where the flux $F_{c_{ij}}$ now has the form

$$
\begin{aligned}
F_{c_{ij}} &= \rho_{ij} \times ([cAMP^j] - [cAMP^i]) \\
&= \omega_{ij} \left[ k_{gj} + \frac{k_{gj,gjf}\bar{P}}{X_{f_3}} \left( \frac{[GJC^i_a]}{P_i} + \frac{[GJC^j_a]}{P_j} \right) \right] ([cAMP^j] - [cAMP^i])
\end{aligned}
\qquad (4)
$$

where $\omega_{ij} = 1$ when cells $i$ and $j$ interface, otherwise $\omega_{ij} = 0$. Here $[GJC^i_a]$ is the active concentration initiated by the delayed regulation model in cell $i$ and likewise $[GJC^j_a]$ in cell $j$. $P_i$ is the

number of cells that cell $i$ couples to and $P_j$ is the number of cells that cell $j$ couples to. Within each cell, the active concentration is parsed equally to each cell it couples to, hence $\frac{[GJC_a^i]}{P_i} + \frac{[GJC_a^j]}{P_j}$ are the parsed concentrations from cells $i$ and $j$, on the $ij$ interface. Thus, $\rho_{ij}$ is time-dependent and modulated by cAMP/PKA activity. The term $\bar{P}$ is the average number of cells that the cells couple to. In this work we set $\bar{P} = 5$, where most $P_i$ typically vary between 4 and 7. $\bar{P}$ and $X_{f_3}$ are normalizing constants so that $k_{gj,gjf}$ has the same dimensions as the basal term $k_{gj}$. For further details, including the ordinary differential equations (ODEs) of the delayed regulation model, the cell-cell coupling flux model $F_{c_{ij}}$, and all molecular species of the multicellular model, please see S4 Text. To reiterate, with this approach, our multicellular model calculates the time-dependent active gap-junction concentrations, and hence the time-dependent effective gap junctional cAMP permeability at all cell-cell interfaces. We are therefore able to calculate all time-dependent cAMP fluxes between cells through the gap-junctions.

Implementing this approach, we find the simple intracellular model with delayed gap-junction regulation is able to capture both the strong overshoot of the ERK-KTR N/C response in the emitter and the simple response with no overshoot in the adjacent receivers (Fig 5I, $N = 4$ and $\tau_{gj} = 15$, for both the fitted ERK-KTR models ($\kappa_{erk} = 1$ and $\kappa_{erk} = .5$ (slowest))). Corresponding dynamics of the active gap junctions on the emitter/receiver interfaces (averaged) is also presented in Fig 5I with a breakdown of the contributions from the emitter and the average of the receivers. Furthermore, adjusting $N$ and $\tau_{gj}$ modulates the overshoot behavior (S7 (A) Fig). And decreasing the number of total receiver cells the single emitter couples to, from 45 cells (Fig 5F) to 10 cells (S7(C) Fig), still results in emitter overshoot while marginally increasing the amplitude of the receiver signal (S7(C) Fig).

We found experimental evidence that supports our delayed gap-junction regulation model. In particular, studies of fluorescent dye flux through gap junctions [46] show that upon titrating in 8-bromo-cAMP, cells exhibit both an increase of flux through connexin43 gap junctions (Figs 1 and 2 from [46], where connexin43 gap junctions are the most efficient gap junction transporters of cAMP [47]) and an increase in membrane gap-junction plaques by 15 minutes (Fig 6 from [46]). This delay in the increase of plaques, which represent regions of active coupling channels, and increased flux is consistent with our cAMP/PKA-dependent, delayed gap-junction regulation model.

## Model predictions and experimental validation

Our model predicts that we will observe similar ERK-KTR N/C overshoot behavior in the presence of the PDE inhibitor IBMX (S8(A) Fig, right panel) as we did without (Fig 5I), and with similar active gap junction molecule dynamics (compare S9(A) Fig (IBMX) with S9(B) Fig (no drug)). The result holds over a range of parameters (S7(B) Fig). We confirmed this experimentally for a small emitter cluster in the presence of the PDE inhibitor IBMX (S8(A) Fig, left panel).

Likewise, the model predicts for the small-emitter-cluster case that a reduction of available gap-junctions to couple to receiver cells will make the emitter signals look more like the all-emitter case (S8(B) Fig, lower right panel (model)). This makes sense since less cAMP flux is entering receiver cells. Indeed this is the case. For a receiver cell that has over expressed connexin-43 with a GFP tagged to its N-terminus (C43-NGFP) [48], these connexins can form gap-junctions at the membrane with other cells, but cannot open [49]. We denote these receiver cells as receiver-CX43-NGFP cells. And its connexin population therefore reduces the level of endogenous working gap-junction channels, thereby reducing flux (upper illustrations in S8(B) Fig). Thus, emitters in a small emitter cluster coupled to receiver-CX43-NGFP cells

cause the emitter cells to behave more like emitters in the all-emitter case (S8(B) Fig, lower left panel), with much less overshoot in the ERK-KTR N/C signal.

Furthermore, for small emitter clusters coupled to receiver cells, if we add an inhibitor of gap junction molecule (connexin) trafficking from the ER to the golgi (Brefeldin A), after 4 hours we should see less overshoot in the ERK-KTR N/C signal in the emitters as well as higher amplitude. This is because connexin trafficking from the golgi to the membrane is greatly reduced [50] because of a lack of connexins in the golgi. In addition, membrane gap-junctions get endocytosed thus depleting the existing membrane gap-junction population as time proceeds. Indeed we see reduced overshoot and increased ERK-KTR amplitude when bPAC is on (S8(C) Fig), with low levels of coupling to receivers.

Overall, we have been able to predict with our model and experimentally confirm how different chemical and genetic perturbations to cell-cell coupling affect the ERK-KTR N/C signal in emitters and adjacent receivers.

### Strong heterogeneity in ERK-KTR N/C response in emitters of single-emitter-cluster experiments indicates a dynamically changing gap junction state

For emitters in an all-emitter experiment, our variability analysis (see S2 Text) shows that the ERK-KTR N/C response tends to be very similar within a given emitter across bPAC input pulses of the same amplitude (single cell responses in S2(A) Fig, for example). In addition, emitters in the all-emitter experiments generally exhibit a very simple first-order like behavior in the responses. However, from our single-emitter-cluster experiment in Fig 3E, it is interesting that there is such a striking difference in the single emitter response across bPAC input pulses of the same amplitude. Indeed, for single-emitter-cluster experiments driven by bPAC input pulses of the same amplitude, we identified that the single emitters tend to exhibit a much larger variation in the dynamics of the ERK-KTR N/C response to bPAC both between different experiments and across different pulses within a given experiment (Fig 6A and S3(A) Fig (right panel)). For example, some overshoots are faster or have higher amplitudes than others, or there may be no overshoot during one of the pulses or both. Furthermore, while our pooled single-emitter-cluster experiments show lower time-to-max statistic for single emitters, on average, than emitters in the all-emitter experiments (Fig 3D, left panel), there are numerous outliers that are much higher (representative example in Fig 6A, second emitter plot from bottom, response during first pulse).

To model this observed variability, we altered our existing model of cAMP/PKA regulated gap-junction activity. Our initial model had particular initial levels of membrane bound gap-junctions and levels of gap-junctions in the recycling/trafficking pathway (see 'far population' in Fig 6B representing inactive gap-junction molecules (connexins) near the nucleus/golgi that must be transported resulting in a large delay ($\sim$ 10 to 15 min.) before activation on the membrane). To model the variability observed in the ERK-KTR N/C response across cells or different pulses within a cell, we add another population, a near population, representing inactive gap-junction molecules (connexins) at or near the plasma membrane that has a smaller activation delay than the far golgi-located population (see 'near population' in Fig 6B for illustration). One could imagine a continuum of populations, each with a different delay, but for this analysis we will consider only two.

Variability in the modeled cAMP and ERK-KTR N/C signals are obtained by adjusting the initial levels of these 2 populations of inactive gap-junctions (see Fig 6B for illustration) for each pulse within a cell. The motivation here is that the recycling/trafficking pathway, during a pulse, alters these populations as well as the active population on the membrane. And the

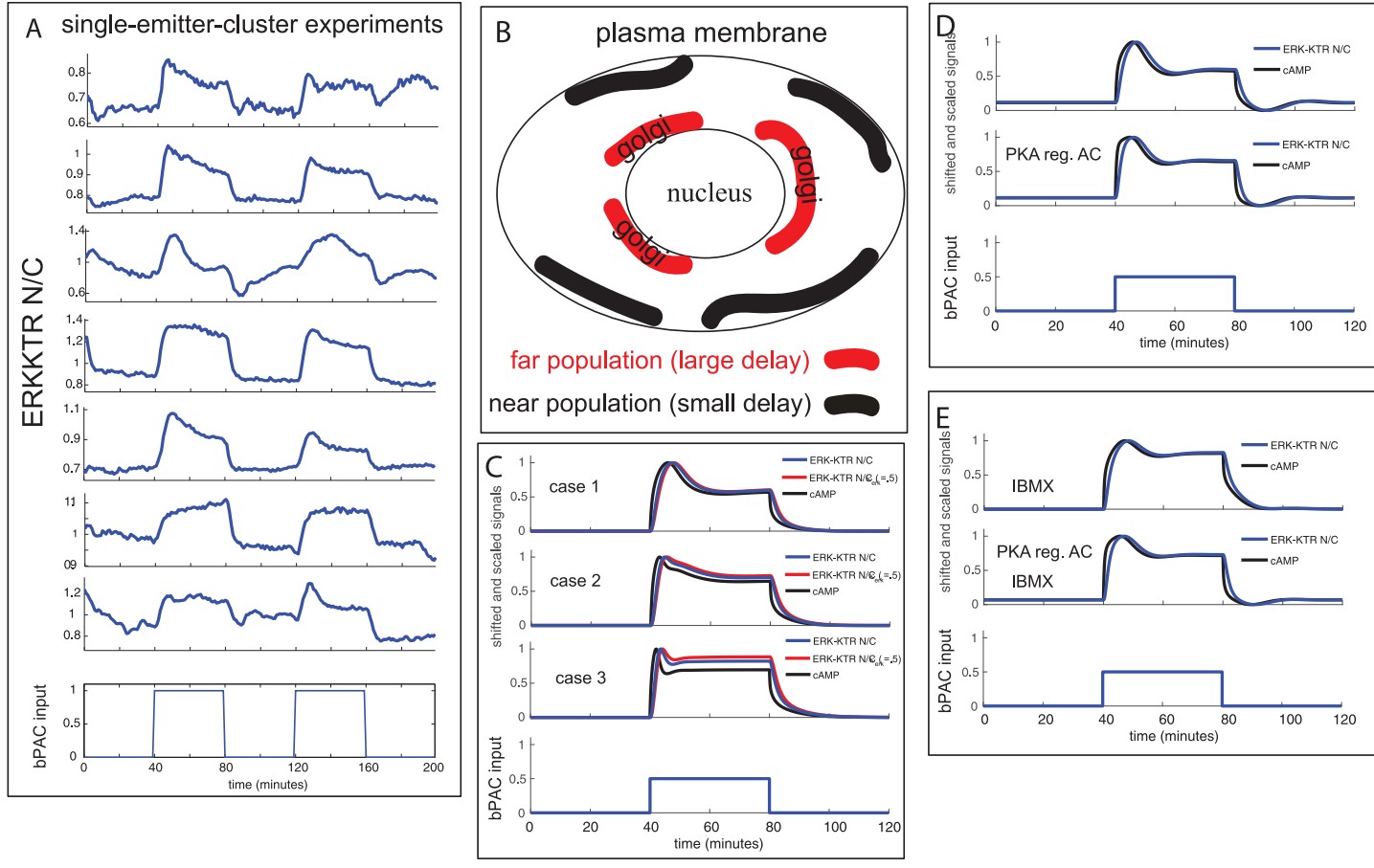

**Fig 6. Differences in the emitter ERK-KTR N/C response across bPAC input pulses of the same amplitude reveal the influence of a dynamically changing gap junction state in single-emitter-cluster experiments.** (A) Single cell ERK-KTR N/C signals from emitters of different single-emitter-cluster experiments all driven by two-pulse bPAC inputs of the same amplitude. (B) Illustration of the fast gap-junction population (black, near the plasma membrane) and the slow gap-junction population (red, closer to the nucleus). (C) Modeling of gap-junction regulation heterogeneity: Case 1 (far only), Case 2 (near + far), Case 3 (near only). Results are presented for both of the fitted ERK-KTR models ($\kappa_{erk} = 1$ (blue) and $\kappa_{erk} = .5$ (red, slowest), as was done in Fig 5I). (D) Modeling the undershoot observed in some single emitters. Top plot: gap-junction regulation only. Middle plot: gap-junction regulation and endogenous adenylyl cyclase feedback. Bottom plot: bPAC input. (E) Modeling the undershoot observed in some single emitters in the presence of PDE inhibitor (IBMX). Top plot: gap-junction regulation only. Middle plot: gap-junction regulation and endogenous adenylyl cyclase feedback. Bottom plot: bPAC input.

populations could further get altered by the recycling/trafficking pathway in between pulses. Thus, at the beginning of the next pulse, the cell will likely have different initial conditions than it had for the prior pulse. For our initial simulations, the near population gets up-regulated within 1 minute or so (parameters: $\tau_{gj} = 1$, and $N = 4$), while the far population gets up-regulated strongly by 15 minutes (parameters: $\tau_{gj} = 15$, and $N = 4$). For all simulation runs the total initial amount of inactive gap-junction molecules we set at $N_i$, where $N_{i_n} + N_{i_f} = N_i$, i.e. the sum of the two populations equals $N_i$. Depending on the amount in each population, we can obtain different cAMP response overshoot signatures, the far only case (Fig 6C, case 1, $N_{i_f} = N_i$) being similar to the data from Fig 6A, third emitter plot from top (first pulse) and bottom emitter plot (second pulse). The near + far modeled response (Fig 6C, case 2, $N_{i_f} = .8N_i$) and near only modeled response (Fig 6C, case 3, $N_{i_f} = 0$) can be seen in experimental data in the top emitter plot of Fig 6C with a 'near + far' type response during the first pulse and a 'near' type response during the second pulse. Results in cases 1–3 are presented in

Fig 6C for both of the fitted ERK-KTR models ($\kappa_{erk} = 1$ (blue) and $\kappa_{erk} = .5$ (red, slowest), as was done in Fig 5I). The corresponding active gap-junction activity at the emitter/receivers interface for cases 1–3 are presented in S9(B)–S9(D) Fig.

These experimental and modeling results suggest a dynamically changing gap-junction state, i.e., a dynamic gap junctional cAMP permeability, in single emitters across pulses that is revealed most strongly in single-emitter-cluster experiments relative to the all-emitter experiments. Importantly, we also see evidence of approximate constant gap junction activity (Fig 6A, second emitter plot from bottom) where, especially for the second pulse, the response has no overshoot and is flat similar to our model of constant gap-junction activity (Fig 5G). And during the first pulse there is a slow continuous rise which could be due to a net loss of gap-junctions over time through endocytosis.

Going back to the single-emitter-cluster example from Fig 3E, one of the receiver cells undergoes mitosis a little before the second bPAC input pulse begins. It has been observed in endothelial cells that during mitosis, gap junctions are removed from the membranes and are intracellular, leading to a loss of gap junction intercellular communication with surrounding cells [51]. Surrounding cells that are non-mitotic keep their gap junctions at cell-cell interfaces and still communicate. Since this loss of gap junction function is likely happening with the receiver cell undergoing mitosis (Fig 3E, cell 37), it would no longer receive cAMP from the emitter cell. Thus, the overshoot in the ERK-KTR N/C signal during the second bPAC input pulse would be due to the remodeling of the gap-junction interfaces in the adjacent non-mitotic receivers and the emitter.

All of these different single cell results point to a picture of the gap-junction state being highly dynamic and heterogeneous between different single-emitter-cluster experiments and across time within a single emitter.

## Discussion

In this work, we applied the optogenetically-controlled adenylyl cyclase bPAC to perturb cAMP levels and observe its effects on ERK signaling through the ERK-KTR N/C signal which is downstream of the cAMP signal and highly sensitive to its dynamics. With our signal analysis and discovery based approach we aimed to decipher the differences in the ERK-KTR N/C signal dynamics that we observe between small-emitter-cluster experiments and all-emitter experiments. In particular, we observed a dynamic overshoot of the ERK-KTR N/C signal in single emitters from small and single emitter cluster experiments that is typically not observed from emitters in all-emitter experiments. Gap junctions were implicated as the source of the dynamic overshoot. This led to the development of a multicellular model of ERK-KTR N/C signal dynamics. And to account for the observed overshoot in single emitter clusters, the model required a delayed gap-junction regulation model of the effective gap junctional cAMP permeability driven by cAMP/PKA. These results are also supported by mechanisms found in the literature [46, 47]. Our experimental system allows us to observe how the EKR-KTR N/C signal in single-emitter-cluster experiments can change drastically across bPAC input pulses of the same amplitude within a given emitter. This suggests a highly dynamic gap-junction state on the cell membrane being driven by gap junction (connexin) trafficking over time which is captured by our computational model based on different initial gap-junction (connexin) conditions prior to each pulse.

While our signal analysis focused on the presence of overshoot in the emitter ERK-KTR N/C responses for both the small-emitter-cluster and single-emitter-cluster experiments, we would like to also note that some of the overshooting emitters exhibit a similar timescale undershoot after bPAC turn-off. This includes some of the emitters from a small emitter

cluster (Fig 3A, single cells), as well as for the emitter of a single emitter cluster during the first pulse (Fig 3E), but not the second. Thus, just as there is heterogeneity in the overshoot observed, there is also heterogeneity in the ERK-KTR N/C signal after bPAC turn-off. In addition, the undershoot occurs with or without PDE inhibition (IBMX) (see emitter ERK-KTR N/C signal undershoot in Fig 3A (wild type) and S8(A) Fig (IBMX), for example), thus, ruling out PDE regulation. The most likely culprit is gap-junction regulation since for small emitter clusters surrounded by receivers that have no gap-junction, we have never observed significant dynamic undershoot. In addition to gap-junction regulation, negative feedback through endogenous adenylyl cyclase might help facilitate the undershoot.

Based on our observations and hypotheses, we adjusted our computational model to account for the undershoot. We found that models with gap-junction regulation only and those that combined gap-junction regulation and feedback on endogenous adenylyl cyclase were both able to produce undershoot (Fig 6D and 6E, with and without IBMX, delay model parameters: $\tau_{gj}$ = 10, and $N$ = 7) for both the ERK-KTR N/C signal and cAMP signal, which behave similarly (see example of corresponding gap-junction activity at the emitter/receivers interface in S9(E) Fig (no IBMX case)). Therefore, at minimum, our model requires gap-junction regulation to produce undershoot when the bPAC input pulse shuts off. It also requires a basal level of cAMP that is higher in emitters than receivers, via higher endogenous adenylyl cyclase activity. For this scenario, during bPAC shutoff, the emitter cAMP level can fall below its basal cAMP level until it is at the basal cAMP level of the surrounding receivers due to strong gap-junction permeability. When the gap-junction permeability decreases due to low cAMP levels, the emitter's intracellular cAMP concentration can rise back to its basal level, hence, the undershoot. We also observe many emitters having very little undershoot after the bPAC input pulse shuts off, suggesting that these emitters may have very low basal levels of cAMP or similar levels to the surrounding receivers which will not allow for an undershoot. Finally, while our delayed gap-junction regulation model can account for undershoot characteristics, the actual trafficking response after bPAC shutoff may have differences in regulation protocols than when bPAC turns on.

We have shown that all-emitter monolayers tend to have little to no overshoot in single cell emitters. Even though all-emitter monolayers still have gap-junction regulation occurring during bPAC input pulses, the net flux of cAMP between cells tend to be small, relative to the emitter/receiver coupling in small-emitter-cluster experiments. This is because every cell is producing cAMP for all-emitter experiments. However, occasionally an emitter will show more pronounced, overshoot, but this is rare. To this end there is variability in the amount of bPAC in each cell. Thus, if neighboring cells have a high differential in cAMP production and concentrations, then it is logical there could be strong flux in the direction of the cell with lower production and concentration. This could yield some amount of overshoot in the emitter with the higher time-dependent cAMP concentration upon upregulation of gap-junctions. Indeed when we account for bPAC expression variability across cells in our model for an all-emitter monolayer, we can observe some cells exhibiting small amounts of overshoot while on average the population does not (S10(B) Fig). On a side but related note, when we model single-emitter-cluster experiments, we find that overshoot in the emitter ERK-KTR signal is preserved for different bPAC expression levels (S10(A) Fig).

In this work, we applied blue light to induce bPAC. This raise the question as to whether or not the blue light pulse directly affects gap junction communication. This is an important one since it has implications of applying optogenetics to tissue studies, in general. As far as evidence of blue light activating gap junctions, we find no evidence in the literature for this. In addition, we would argue that the small-emitter-cluster experiment with PKA inhibition in Fig 4C supports this. That is, in the presence of blue-light (during the second bPAC input pulse)

and inhibited PKA (via H89), there is very little coupling from the emitter to the receiver as compared to the strong coupling during first pulse prior to addition of H89. A potential interpretation from these results is that if blue light somehow enhances gap junction activity, it would be through enhancing PKA activation, complementing cAMP activation. But, there is no evidence of PKA being light sensitive since we see no ERK-KTR N/C response when we illuminate an all-receiver monolayer with blue light (S1(G) Fig). These results suggest that gap-junction activity is not enhanced by blue light.

Our well-mixed, multicellular model allows us to simulate signal propagation and delay from the emitter to the surrounding receivers (S10(C) Fig). As a function of receiver cell distance from the emitter, the peak ERK-KTR N/C signal decreases and the time to max signal (time-to-max) increases. Of the cells whose time signals are plotted, the model even shows a slight delay in initial rise (flat before initial rise) of the ERK-KTR N/C signal for the cell furthest from the emitter (cell 13). In S3(D) Fig, we showed a single emitter cluster example where the peak receiver ERK-KTR N/C amplitudes drop of as a function of distance from the emitter. Examples of the time signals from this experiment, including the emitter and a few receivers at different distances, are shown in S10(D) Fig (top plots and middle plots). Similar to the model, the further a cell is away from the emitter, the lower the peak tends to be, as well as the longer the time to reach max amplitude (time-to-max). However, in this experiment, the further a receiver is from the emitter, the lower the amplitude is and thus the lower the signal to noise ratio. Thus, trying to assign delay in initial rise (ERK-KTR N/C signal) to these receivers is not quantitative. To better study delays in the initial rise, future experiments of propagation and associated delays could use a larger field-of-view to get more cells, at least double the dimensions. And larger emitter clusters would allow for a higher level of signal strength (higher signal-to-noise ratio) as a function of receiver distance from the emitters.

The main focus of this work has been dissecting regulations due to the effects of cell-cell coupling on intracellular signaling. However, in addition, we have found evidence of a potential bPAC-induced emergent behavior within the receiver cells from small-emitter-cluster experiments, specifically a mobility phenotype. It occurs when we perturb cAMP levels in small-emitter-cluster experiments where a subset of receiver cells become mobile and move away from the emitters (examples in S11(A), S11(E) and S11(F) Fig with further discussion in S5 Text and S1 Table (also see S5 Movie with description)). We have not observed it in all-emitter experiments nor small-emitter-cluster experiments when gap-junctions are inhibited, including PKA inhibition (examples in S11(B), S11(C) and S11(D) Fig). The data suggest that the requirements are a cAMP perturbation in a small group of emitters as well as PKA and potentially gap-junctions. In addition, the strongest mobility phenotype occurs for cases when emitters are surrounded by a receiver cell line that form mostly broken gap-junctions thereby reducing the cAMP coupling flux. This potentially points to a gradient in PKA activity between emitters and receivers [52] that might drive the mobility phenotype. Further experimental characterization of this phenotype is a subject of current research.

Overall, our systems level approach with precise spatio-temporal input control and pathway measurements serves to quantitatively elucidate cell-cell coordination and decision making that give rise to context-dependent tissue function and response. While for this work, we have focused on the singular perturbing of cAMP, we envision future studies to employ combinatorial perturbations of various signalling molecules and external environmental signals such as EGF for quantitative studies including how cross-talk between pathways affects their dynamics and decision making. Furthermore, our work lays the foundation as a general approach for studying the differences in cell-cell communication and coordination between healthy and diseased in-vitro mosaic tissues.

## Materials and methods

### Cell culture and reagents

MDCKI and MDCKII cell lines were a kind gift from Pavel Nedvestky. No mycoplasma contamination was found. MDCKI + bPAC cells were generated by infecting MDCKI cells with pLenti-PGK- bPAC::NES (Addgene #130267). ERK-KTR expressing cells were obtained by transfecting MDCKI and MDCKI + bPAC cells with either pPB-CAG-RFP-ERKKTR or pPB-YFP-ERKKTR. PKAc (PRKACA reporter) expressing cells were obtained by transfecting MDCKI and MDCKI + bPAC cells with pPB-CAG-PRKACA::mRuby2 (Addgene #130268).

Overexpressed CX43-NGFP (broken connexin43) expressing cells were obtained by transfecting MDCKI and MDCKI + bPAC cells with sfGFP-Cx43K258stop (Addgene #69025) or msfGFP-Cx43 (Addgene: #69024).

Plasmids for the overexpressed CX43-NGFP (broken connexin43) expressing cells were constructed using the Mammalian Toolkit (MTK) [53], a hierarchical DNA assembly method based on Golden-Gate (GG) cloning [54]. New binding domains were domesticated into the MTK by ordering primers or gBlocks from IDT that contained the desired coding sequence with all BsaI and BsmBI restriction sites replaced with synonymous mutations and overhangs for MTK Part 3as, then ligated into the MTK part entry vector using a BsmBI GG reaction. The plasmids were added then to MDCKI + bPAC + RFP-ERKKTR and MDCKI + RFP-ERKKTR cells for experiments when ERK-KTR measurements were also required.

Cells were maintained in MEM (Gibco #11095072) supplemented with 10 percent Fetal Calf Serum (VWR #89510–184) and 1X Anti-Anti (Gibco #15240062), and kept at 37 degrees (C) in a humidified incubator with 5 percent CO2.

For activators and inhibitor experiments, small molecules and doxycycline were dissolved to typically 4X to 10X greater than the IC50 concentration (EC50 for activation) in imaging media before addition. The aim was for the effect of the drug to be maximized (either inhibition or activition) while minimizing stress and other physiological effects, when possible. Final DMSO concentration did not exceed 0.15%. Inhibitors included Epac inhibitor ESI-09 (15 $\mu$M, ESI-09 IC50 inhibition of Epac1 at 2.4 $\mu$M and Epac2 at 4.4 $\mu$M, in presence of 20 $\mu$M cAMP (competitive binder)), PKA inhibitor H89 (10 $\mu$M, IC50 of .135 $\mu$M), PDE inibitor IBMX (100 $\mu$M, IC50 values are 13, 18, 19, 32 and 50 $\mu$M for PDE4, PDE3, PDE1, PDE5 and PDE2 respectively), Gap junction inhibitor carbenoxolone (10 $\mu$M), ER to GolgI trafficking inhibitor Brefeldin A (10 $\mu$M). Activators included the adenalyl cyclase activator Forskolin (50 $\mu$M, saturating). All inhibitors and activators were purchased from Tocris.

### Microscopy

To prep cells for continuous experiments (including those when drugs were added prior to imaging), cells were plated in Greiner Bio-One 4-compartment dishes (GBO #627870) where each compartment received 1 mL of cells/media ($2.5^{*}10^{5}$ cells/mL with media phenol-red-free MEM (Gibco #51200038) with 2% Fetal Calf Serum (VWR #89510–184)). To prep cells for experiments when drugs were added during imaging, cells were plated in Ibidi $\mu$-Slide 8 Well Grid-500 (Ibidi #80826-G500, $.9^{*}10^{5}$ cells/mL, .3 mL of cells/media per well). This prep was also used for experiments involving the PRKACA reporter that quantifies PKA activity. For any of the above experiments, the plated cells were then incubated for at least 16 hours prior to imaging. An hour before imaging, the cells were stained with DNA binding dyes (Hoechst) as a nuclear marker (DAPI channel). The same media phenol-red-free MEM with 2% Fetal Calf Serum was replenished and if any small molecule was required at the beginning of the experiment, it was added with the replenished media. Imaging would begin at least an hour later.

Widefield microscopy images were taken on a Nikon Ti-E inverted scope, with an incubation enclosure set to 37 C (Ibidi Temperature controller), 5.0 percent CO2 and 80 percent humuidity (Ibidi CO2 and humidity controller), and mercury arc-lamp illumination using RFP filters and dichroic mirror from the Chroma 89006 ET-ECFP/EYFP/mCherry filter set. The microscope's perfect focus system was used to maintain focus on the samples throughout the experiment. The microscope was controlled with micro-manager software version 1.4.17 using the High Content Screening Site Generator Plugin (Edelstein et al., 2014). We used custom MATLAB scripts to control micro-manager and blue LED/Halogen lamps with a neutral density filter (max light intensity of $F_{max} \approx 200 \; \mu\text{W cm}^{-2}$ where the EC50 of bPAC activation saturation is 370 $\mu\text{W cm}^{-2}$ [22].) to induce emitter cells (cells containing bPAC) with user-defined blue light inputs. Note that for the 6-minute-pulse experiments in S1(B) Fig, the neutral density filter was not used and $\alpha$ represents the resulting scaled increase in photon flux where $\alpha \approx 7$. Images were taken with an Andor 512 pixel EMCCD camera (897 iAxone DU-897E) using a 40x objective (Nikon Planfluor 40x/0.75) and 1.5x optical zoom. This resulted in a typical range of 40 to 80 cells in the field of view during acquisition. At 40x, the camera resolution is .4 microns per pixel (example distance calculation in S3(D) Fig).

Key conditions from imaging experiments were performed at least twice, with one replicate presented in figures.

## Choice of 40 minute bPAC input pulses

For the majority of our experiments we use 40 minute bPAC inputs pulses. The 40 minute duration was chosen because the majority of cells reach steady state in their ERK-KTR N/C signal by that time. Thus, we can observe the single-cell dynamic response to completion. Twenty minute pulses are too short where the ERK-KTR N/C signal can still be exhibiting dynamics by twenty minutes. In addition, we've done 8 hour experiments on single monolayers with duty cycle 40 minute on/40 minute off, and the cells appear healthy during the whole experiment. And for a similar reason we chose an off time of 40 minutes so the cell tend to be at an approximate steady-state when the next bPAC input pulse starts.

## Image analysis and signal extraction and quantification of pathway activity reporters

We used custom MATLAB nuclear segmentation scripts that apply adaptive thresholding approaches [55] on the sequence of DAPI images obtained which results in a group of objects, some single nuclei and some clustered/overlapping nuclei (typically 2–5 nuclei). A recursive adaptive thresholding algorithm with a minimum nuclear size constraint (200 pixels for our 40X images) is then applied to each individual object in order to further separate clusters into individual nuclei. For each individual segmented nuclei object obtained, a threshold of half the peak nuclear DAPI intensity is applied to that nuclei in the DAPI image to obtain the final segmented object for the individual nuclei. Any remaining clusted/overlapping nuclei are discarded in further data processing and analysis. Our segmentation protocol identifies nuclei and its nuclei position (center of mass) for each image. Nuclei are tracked across images based on determining the closest individual nuclei between images, allowing for the tracking of individual nuclei over time. DAPI images are obtained every three minutes, a sufficient sampling period to minimize errors in individual nuclei assignment across images.

Emitter nuclei also have the genetic markers (tagBFP-nls or Lamin B1-mTFP1) in addition to the DNA binding dyes (Hoechst). For experiments containing emitters and receivers, the MATLAB scripts scan for this genetic markers (above a minimum threshold) within the segmented nuclei to determine the nuclei of emitter cells and receiver cells within the images.

With each nuclei identified, we determine the associated cytoplasmic region of each cell by following the established procedure of creating a ring around each nucleus with a half-nuclear width [32]. We then identify and track individual cells over time by mapping their nuclear locations and corresponding reporter readout from frame to frame. For each cell in a monolayer, we can therefore extract its spatial coordinates and the time dependent profile of the nuclear and cytoplasmic magnitude of its various reporters.

In order to measure ERK activity in MDCK cells, we have implemented an ERK responsive Kinase Translocation Reporters (KTR) [32]. Here, phosphorylation by ERK is transformed into a nucleocytoplasmic shuttling event of the KTR that can be measured by fluorescence microscopy. In this technology, kinase activity leads to phosphorylation of the Nuclear Localization Sequence—Nuclear Export Sequence of the designed reporter, and hence increases its cytoplasmic residence. Conversely, reduced kinase activity leads to accumulation of reporter in the nucleus. We successfully expressed a KTR for ERK (ERKKTR) fused to either Clover or tagRFP fluorescent proteins in MDCK cells. For each tracked cell, the ratio of the nuclear median signal over the cytosolic median signal was calculated for each frame. Cells that did not respond over the duration of the experiments were discarded when calculating the population averaged signal and population standard deviation of the signal for a given experiment.

In order to measure PKA activity in MDCK cells we have expressed Venus or tagRFP fused to PRKACA, one of the catalytic subunits of PKA [11]. Upon binding to cAMP, the regulatory subunits of PKA disengage from the catalytic subunits, allowing them to change their cellular distribution. This process is easily measured with the change from cytoplasmic puncta to whole cell diffuse fluorescence. Additionally, we tagged the endogenous PRKACA protein with Venus, in order to avoid the impact of extra copies of PKA catalytic subunits.

To measure the quantitative dynamics of intracellular $Ca^{2+}$, we have expressed the genetically encoded calcium indicators (GECIs) GCaMP5g and RCaMP1h. GCaMP5 is based on a circularly permuted green fluorescent protein (cpGFP), calmodulin (CaM), and a $Ca^{2+}$/CaM-binding (M13) peptide. RCaMP1h, on the other hand, is engineered from circular permutation of the thermostable red fluorescent protein mRuby. Upon binding to calcium, these molecules change their conformation, and by doing so, increase their fluorescence. Both sensors have high sensitivity and rapid dynamics [37, 38].

## Statistical methods

We applied the student t test for paired samples (single tailed) to show that time-to-max measurements of the ERK-KTR N/C signal from one group of pooled experiments has a greater mean than that of another group of pooled experiments. We also applied the student t test for paired samples (single tailed) in the same manner for overshoot ratio measurements. For each of the pooled data sets, we used the measurements from each of the two bPAC input pulses for each single cell. The p-values are reported in the main text where the means are compared between data sets.

## Computational model

The computational model representing the multicellular cAMP signaling network is described in S4 Text. Model simulations were performed with MATLAB ordinary differential equation solver ODE15S. Parameter values for the model simulations are presented in S2 and S3 Tables.

## Dryad DOI

Data from imaging experiments was deposited at the Dryad Digital Repository: https://doi.org/10.7272/Q6QC01RN [56].

## Supporting information

**S1 Fig. The ERK-KTR reporter is sensitive to cAMP dynamics.** (A) Activation of bPAC results in higher firing rate of $Ca^{2+}$ spikes in single cells. (B-C) ERK-KTR N/C signal due to bPAC input pules of increasing amplitude (maximum amplitude of one corresponds to the maximum blue light photon flux of the system with a neutral density filter). Left panel: Top plot: emitter population ERK-KTR N/C signal average. Middle plots: single cell ERK-KTR N/C signals. Green segments on the ERK-KTR N/C signal show when the bPAC pulse is on. Bottom plot: bPAC input pulse sequence. Right panel: ΔERK-KTR N/C versus bPAC pulse amplitude. ΔERK-KTR N/C is the difference between the ERK-KTR N/C signal at the start of the pulse and the signal at the specified time of measurement. (B) ERK-KTR N/C signal due to 6 minute bPAC input pulses with increasing amplitudes of $.2\alpha$, $.4\alpha$, $.6\alpha$, and $1\alpha$ for all-emitter monolayers for cells with phosphodiesterase (PDE) inhibition (IBMX, red) and with no IBMX (blue). For these experiments a neutral density filter was not used and $\alpha$ represents the resulting scaled increase in photon flux where $\alpha \approx 7$. ΔERK-KTR measurements are taken at 2 minutes and 6 minutes after the start of the pulse (see black (2m) and blue (6m) stars in first pulse of the top single cell plot, for example). (C) ERK-KTR N/C signal due to 40 minute bPAC input pulses with increasing amplitudes of amplitudes of .125 and .25. ΔERK-KTR measurements are taken at the end of each pulse (see top single cell plot). (D) ERK-KTR N/C signal due to 40 minute bPAC input steps with increasing amplitudes of .2 and .4 for an all-emitter monolayer. (E) Single cell exhibiting saturation in its ERK-KTR N/C signal (top plot) with a large delay before decaying after the bPAC input pulse (bottom plot) has shut off. Compare with a single cell that is within the dynamic range of the reporter (middle plot) and with a decay delay on the order of a minute or two, the natural delay of the reporter when changes in cAMP occur. (F) Example of single cell ERK-KTR N/C signal that exhibits proportional slope in the rise versus bPAC input pulse amplitude for a pulse sequence of increasing amplitudes. However, max ERK-KTR N/C signal is not proportional versus bPAC input pulse amplitude. Some saturation occurs with noticeable decay delay after the second bPAC input pulse shuts off. (G) All-receiver monolayer exhibits no ERK-KTR response when illuminated by blue light.
(PDF)

**S2 Fig. Variability analysis of ERK-KTR N/C signal for individual all-emitter experiments.** (A-D) Calculation of the population mean and standard deviation of the ERK-KTR N/C signal under different normalization approaches (same approach as in Fig 2E and 2F). Top 3 plots: For a given normalization approach, each single cell ERK-KTR N/C signal is shifted such that the signal at the beginning of the first pulse is zero and then normalized. Mean and standard deviation are calculated to just before the start of the second pulse. This process is the then repeated for the second pulse enforcing the standard deviation is zero at the beginning of each pulse. Top plot: mean ± standard deviation. Second plot from top: mean ± standard deviation calculated after each shifted signal is normalized by difference between the peak value of the signal during the first pulse and the value at the beginning of the first pulse. This removes the effects of peak amplitude variability in the first pulse. Third plot from top: same as prior normalization approach of the first pulse, but applied to the second pulse instead. This removes the effects of peak amplitude variability in the second pulse. Second, third, and fourth plots from the bottom: single cell ERK-KTR N/C signals. Bottom plot: bPAC input pulse sequence.

(A) All-emitter experiment with identical bPAC input pulse amplitudes. (B) All-emitter experiment with increasing bPAC input pulse amplitudes. (C) All-emitter experiment with increasing bPAC input pulse amplitudes with PKA inhibition. (D) All-emitter experiment with increasing bPAC input pulse amplitudes with PKA inhibition. (E) Scatter plot of time-to-max times (same approach as described in Fig 3C) for single cell ERK-KTR N/C signals taken from all-emitter experiments from (A), (B), and Fig 2E and 2F, and all-emitter experiments with PKA inhibition from (C) and (D). Mean (magenta diamond) ± standard deviation (magenta error bars), error bars in the mean (green, standard deviation/$\sqrt{N}$) representing 95 percent confidence interval in the mean, and median (yellow diamond) are also plotted. Number of data points $N$ used are displayed below each group label.
(PDF)

**S3 Fig. Measurements from various small-emitter-cluster and single-emitter-cluster experiments.** A) Heterogeneity is observed in emitter response for small emitter clusters and single emitter clusters. Left two panels: Example plots of emitter ERK-KTR N/C signals from two small emitter clusters of size $N = 5$ and 6, respectively. Bottom plot: bPAC input. Right panel: Examples of emitter ERK-KTR N/C signals from different single emitter clusters. Bottom plot: bPAC input. (B) Examples of single emitter and adjacent receiver average ERK-KTR N/C signals along with time snapshots of emitter ERK-KTR response (within outline) displaying the change in nuclear ERK-KTR after overshoot. (C) Scatter plot of time-to-ss—time-to-max statistics for the small-emitter-cluster experiments (green, with left most from experiment presented in Fig 3A), and combined data for 10 single-emitter-cluster experiments, all presented Fig 3D. Mean (magenta diamond) ± standard deviation (magenta error bars), error bars in the mean (green, standard deviation/$\sqrt{N}$) representing 95 percent confidence interval in the mean, and median (yellow diamond) are also plotted. Number of data points $N$ used are displayed below each group label. (D) Single emitter experiment that shows a general decreasing trend of signal amplitude vs. distance from the emitters. Left Panel: Image of single emitter (purple nucleus, cyan label), adjacent receivers (red nuclei, magenta label) and all other receivers (red nuclei, yellow label). Cyan circle centered at emitter represents a radius of 80 $\mu$m (.4 $\mu$m per pixel). Right Panel: Normalized steady-state signal amplitude (relative to emitter) of receiver cells as a function of distance from emitter (cyan square at zero distance). Steady-state signals are measured at the end of the bPAC input pulse. Adjacent receiver cells from image are labeled as magenta diamonds. All other receiver cells are labeled as green circles. Cells that did not respond to bPAC were not included (including adjacent receivers 34 and 48).
(PDF)

**S4 Fig. Measurements from various small-emitter-cluster and single-emitter-cluster experiments under various inhibitions of gap-junctions.** (A) Comparison of ERK-KTR N/C signal decay after bPAC input pulse shutoff. In red (top plot) is the average emitter ERK-KTR N/C signal for the small-emitter-cluster experiment with gap-junction inhibition (carbenoxolone) from Fig 4A. Plotted for comparison (blue) is the average emitter ERK-KTR N/C signal for an all-emitter experiment (no chemical input). Signals are shifted and scaled to have same peak amplitudes in order to be able to directly compare signal decay after the bPAC input pulse (bottom plot) turns off. (B-C) Plots of mean and standard deviation for small-emitter-cluster experiments under different chemical inhibitions. For standard deviation calculations, each single cell ERK-KTR N/C signal is shifted such that the signal at the beginning of the first pulse is zero to remove baseline variability. Standard deviation is calculated to just before the start of the second pulse. This process is the then repeated for the second pulse enforcing the standard deviation is zero at the beginning of each pulse. This is done for both the emitter groups and

the receiver groups. Top plot: emitters, Middle plot: adjacent receivers, Bottom plot: bPAC input. (B) Gap junction inhibition (carbenoxolone) experiment from Fig 4A. (C) PKA inhibition (H89) experiment from Fig 4B. (D-E) Experimental results for small and single emitter clusters surrounded by receivers that do not form gap junctions (MDCKII cells). (D) Average emitter ERK-KTR N/C signal for different small emitter clusters each with different bPAC input pulse sequences. (E) Emitter ERK-KTR N/C signal for different single emitter clusters each with different bPAC input pulse sequences. Image depicts the single emitter cluster (receivers do not have ERK-KTR but have a nuclear marker (red nuclei, yellow label)) whose ERK-KTR N/C signal is plotted in the middle panel.

(PDF)

**S5 Fig. Both Epac and PKA contribute to the intracellular ERK-KTR N/C signal.** (A) Testing how Epac and PKA inhibitors affect ERK under endogeneous adenalyl cyclase activation through Forskolin (all-receiver experiment). Cells are imaged for twenty mintutes. Forskolin is then added, and the same cells are imaged for twenty minutes. An inhibitor(s) is then added and the sames cells are imaged for the final twenty minutes. All panels: top plot: average ERK-KTR N/C signal, middle plots: single cell ERK-KTR N/C signals, bottom plot: step input of Forskolin. Left Panel: Epac inhbitor (ESI-09), Middle Panel: PKA inibition (H89), Right Panel: both inhibitors (ESI-09 and H89). Note that the addition of Forksolin disturbs the microscope where it can take a few minutes to find and realign the same cells, causing a discontinuity in the measured signal. The same holds true for the addition of the inhibitor(s). (B) Testing how Epac and PKA inhibitors affect ERK under bPAC activation (all-emitter experiment). Cells are imaged during a pulsed bPAC input sequence (40 minutes off, 40 minutes on, 40 minutes off). An inhibitor(s) is than added and the bPAC input sequence sequence is repeated while the same cells are imaged. All panels: top plot: average ERK-KTR N/C signal, middle plots: single cell ERK-KTR N/C signals, bottom plot: bPAC input pulse sequence. Left Panel: Epac inhbitor (ESI-09), Middle Panel: PKA inibition (H89), Right Panel: both inhibitors (ESI-09 and H89). Note that the addition of the inhibitor(s) disturbs the microscope where it can take a few minutes to find and realign the same cells, causing a discontinuity in the measured signal. (C) All-emitter monolayer experiment with PKA inhibition (H89): top plot: average ERK-KTR N/C signal of emitters. bottom plot: bPAC input pulse sequence of increasing amplitudes. The average ERK-KTR N/C signal exhibits similar qualitative behavior to the all-emitter experiment (no PKA inhibition) from Fig 2F. (D) Literature-based illustration of multi-cellular model where each cell has an intracellular circuit and couple to other cells through gap-junctions.

(PDF)

**S6 Fig. Model fitting of intracellular ERK-KTR N/C signal, single-emitter-cluster model results with constant gap-junctional permeability, and characterizing the gap junction delay model for different *N* values.** (A-E) Modeling results for simplified intracellular pathway with no feedbacks in an all-emitter monolayer. Model is fitted to ERK-KTR N/C data for 6 minute bPAC input pulses applied to an all-emitter monolayer (S1(B) Fig, IBMX, population average). We present results for two values of the deactivation rate of ERK $\gamma_{erk}$ using the expression $\kappa_{erk}\gamma_{erk}$ where $\kappa_{erk}$ is a scaling factor and $\gamma_{erk}$ is a fitted deactivation rate. ERK-KTR N/C and cAMP signals are shifted to zero and scaled for comparison, with the peak of the emitter signal at 1 and minimum at zero. (A) ERK-KTR N/C signal and predicted cAMP signal driven by 6 minute bPAC input pulse. Upper plot: fitted model ($\kappa_{erk} = 1$). Middle plot: same fit except for a slower ERK deactivation rate reduced by a factor $\kappa_{erk} = .5$. (B) Same as (A) but with IBMX. (C) Same plots as (B) but with the ERK-KTR N/C data overlayed to show the fit for both the $\kappa_{erk} = 1$ and $\kappa_{erk} = .5$ cases. (D) Same as (A) but for 40 minute bPAC input pulses.

(E) Same as (B) but for 40 minute bPAC input pulses with IBMX. (F) PKA signal for 40 minute bPAC input pulses (from same modeling results as (D)). (G-H) ERK-KTR N/C and cAMP signals are shifted to zero and scaled for comparison, with the peak of the emitter signal at 1 and the minimum at zero. The receiver signals are multiplied by the same scaling factor as the emitter to maintain the same relative amplitudes. (G) Single-emitter-cluster modeling results for the model presented in Fig 5G, but where the gap junctional permeability is reduced by a factor of 3 ($\kappa_{gj} = 1/3$) or increased by a factor of 3 ($\kappa_{gj} = 3$). (H) Single-emitter-cluster modeling results for the model presented in Fig 5H, but where the gap junctional permeability is reduced by a factor of 3 ($\kappa_{gj} = 1/3$) or increased by a factor of 3 ($\kappa_{gj} = 3$). (I) All-emitter modeling results for the intracellular circuit with PKA feedback through the PDEs presented in Fig 5H. In top plot, the ERK-KTR N/C and cAMP signals are shifted to zero and scaled for comparison, with the peak of the emitter signal at 1 and the minimum at zero. (J) Simulation results of the $N$-step delay model for $N = 1, 2, 4, 10, \infty$ and delay $\tau_{gj}$. Plotted is the time-dependent active state $GJC_a^i$ given that at $t = 0$, $GJ_{d_1}^i = K$.
(PDF)

**S7 Fig. Parameter exploration of the delayed gap-junction regulation model.** (A-B) Parameter sweep of $\tau_{gj} = 7, 15, 20$ and $N = 1, 4, 7$, where $\tau_{gj} = 15$, $N = 4$ represents the values used for the results in Fig 5I. ERK-KTR N/C and cAMP signals are shifted to zero and scaled for comparison, with the peak of the emitter signal at 1 and the minimum at zero. The receiver signals are multiplied by the same scaling factor as the emitter to maintain the same relative amplitudes. (A) without IBMX. (B) with IBMX. (C) Comparison of model results presented in Fig 5I, which simulates 46 cells, to simulation results that use 11 cells represented in left panel (emitter (yellow nucleus, red label) and receiver cells (light green nuclei, black label), and where lines connecting nuclei represent direct cell-cell communication through gap-junctions).
(PDF)

**S8 Fig. Experimental confirmations of the delayed gap-junction regulation model predictions through PDE inhibition or experimentally reducing available working gap-junctions.** For (A), (C) and bottom 3 panels in (B): Left Panel: top plot: average small-emitter-cluster ERK-KTR N/C signal, middle plot: average adjacent receivers ERK-KTR N/C signal, bottom plot: bPAC input pulse sequence. Middle Panel: image of small emitter cluster (cluster emitters: white labels, adjacent receivers: yellow labels). Right Panel: predicted model results. (A) Experimental observations and model predictions where an emitter in a small emitter cluster exhibits ERK-KTR N/C overshoot under PDE inhibition (IBMX). Same model used from Fig 5I, but with the PDE dependent cAMP decay rate set to zero. (B) Cell-cell coupling models for emitters coupled to different receiver strains. Top left panel: emitter and receiver strains contain the endogenous connexin43. Top right panel: emitter strain coupled to the receiver-CX43-NGFP strain which contains both the endogenous connexin43 and the overexpressed connexin43-NGFP that can form gap-junctions but no flux. Bottom left panel: ERK-KTR N/C signals for small-emitter-cluster and receiver-CX43-NGFP experiment. Over expression of broken connexin43 (CX43-NGFP) reduces cAMP coupling through competition with endogenous connexin43. Bottom right panel: same model used from Fig 5I, but with the max gap junctional transport rate of cAMP set to .2 that of the wild-type to capture the reduced coupling effect. (C) Emitter in single emitter cluster shows large amplitude with very little overshoot in the ERK-KTR N/C signal in the presence of the trafficking inhibitor Brefeldin-A. A small amount of coupling to receiver cells is observed. Same model used from Fig 5I, but with the max gap junctional transport rate of cAMP set to .25 that of the wild-type to capture the reduced coupling effect.
(PDF)

**S9 Fig. Model results of the delayed gap-junction regulation model.** (A)-(E) Active gap-junction state dynamics at the emitter/receivers interface for the modeled single-emitter-cluster for various cases throughout the paper. Top plot: emitter ERK-KTR N/C signal (shifted and scaled). Second plot from the top: average of adjacent receivers ERK-KTR N/C signal (shifted and scaled by same factor as emitter). Third plot from the top: corresponding cAMP/ PKA driven dynamics of the active gap junctions on the emitter/receiver interfaces (averaged). Also included is a breakdown of the contributions from the emitter and the average of the adjacent receivers. Bottom plot: bPAC input pulse sequence. (A) Model presented in S8(A) Fig. Same model as Fig 5I ($\tau_{gj}$ = 15, $N$ = 4), but in the presence of PDE inibitor IBMX. (B) Model presented in Fig 5I ($\tau_{gj}$ = 15, $N$ = 4) as well as in Fig 6C (case 1, slow population only). (C) Model presented in Fig 6C (case 2, slow and fast population mixture). Fourth plot from the top: a breakdown of the contribution from the emitter into its slow and fast populations. Fifth plot from the top: a breakdown of the contribution from the average of the adjacent receivers into its slow and fast populations. (D) Model presented in Fig 6C (case 3, fast population only). (E) Model which accounts for undershoot of the ERK-KTR N/C signal corresponding to Fig 6D (top plot, no IBMX, no PKA regulation of AC).
(PDF)

**S10 Fig. Examining bPAC expression variability and signal propagation with the multicellular model.** (A) Varying the bPAC expression level for the emitter in the single-emitter-cluster model where [$bPAC_0$] has been the implicit concentration throughout the paper. And where the value of $\beta_{c,b}$ (cAMP production rate due to bPAC) is proportional to the bPAC expression level. Presented are results for expression levels [$bPAC_0$]/2, [$bPAC_0$], and [$bPAC_0$] $\times$ 2. Note that this is effectively the same as scaling a given bPAC input amplitude by the same factor. (B) Simulating cell-cell heterogeneity in bPAC expression across an all-emitter monolayer for different bPAC input pulse amplitudes. Cell-to-cell differences (gradients) in bPAC expression can yield single emitters with small amounts of overshoot (middle plot) relative to the average (top plot). (C)-(D) Top panel: For top plot, ERK-KTR N/C signals are shifted to zero at the beginning of the bPAC input pulse and scaled by the resulting peak emitter amplitude which results in the peak of the emitter signal at 1. The receiver signals, multiplied by the same scaling factor as the emitter, maintain the same relative amplitudes (proportionally). Second plot from top: same as top plot but all plotted signals have their peak normalized to 1 to better examine delays. Bottom plot: bPAC input. Bottom panel: Image of multicellular system from which signals are measured. (C) Model example of ERK-KTR N/C signal propagation from a single-emitter-cluster. Third plot from top: corresponding dynamics of the active gap junctions on the cell-cell interfaces for the cell-cell pairs measured. In bottom panel: emitter (yellow nucleus, red label), receivers (light green nuclei, black label). (D) Experimental example of ERK-KTR N/C signal propagation from a single-emitter-cluster experiment. In bottom panel: emitter (white label), receivers (yellow label).
(PDF)

**S11 Fig. Cell-cell coupling through gap-junctions is potentially required for observed emitter/cAMP driven receiver-mobility phenotype in epithelial monolayers. And the phenotype is potentially regulated through PKA.** (A-F) Illustration depicts overall mobility of cells over the course of an experiment. The image captures the final positions of the nuclei (emitters (white label), receivers (yellow label)) at the end of the experiment overlayed with their individual mobility trajectories that occurred over the duration of the experiment. (A) Small-emitter-cluster experiment (Fig 3A) exhibiting emitter/receiver cAMP coupling over a 2 hour and 40 minute duration. (B) All-emitter experiment (Fig 2B) over a 3 hour and 20 minute duration. (C) Gap-junction inhibition experiment (carbenoxolone, Fig 4A) exhibiting no emitter/

receiver cAMP coupling over a 2 hour duration. (D) PKA inhibition experiment (Fig 4B) exhibiting little to no emitter/receiver cAMP coupling over a 3 hour and 20 minute duration. (E) Small emitter cluster and receiver-C43-NGFP experiment (S8(B) Fig) exhibiting reduced emitter/receiver cAMP coupling over a 3 hour and 20 minute duration. (F) Small emitter-C43-NGFP cluster and receiver experiment exhibiting reduced emitter/receiver cAMP coupling over a 2 hour and 20 minute duration.
(PDF)

**S1 Movie. Description: Movie of all-emitter monolayer with consecutive bPAC input pulses of the same amplitude.** Details of experiment: All emitter cluster with ERK-KTR-YFP and consecutive bPAC input pulses of the amplitude $F_{max}$. Note when making the movie, to raise the visibility of the cells with lower expression of ERK-KTR-YFP, we set a threshold to half the peak fluorescence observed in the images over the course of the experiment. Thus, for a few cells, the ERK-KTR nuclear signal will appear to saturate which is just an artifact of the thresholding.
(AVI)

**S2 Movie. Description: Movie of all-emitter monolayer with consecutive bPAC input pulses of increasing amplitude.** Details of experiment: All emitter cluster with ERK-KTR-YFP and consecutive bPAC input pulse amplitudes of $.125F_{max}$ and $.25F_{max}$, respectively.
(AVI)

**S3 Movie. Description: Movie of small emitter cluster.** Details of experiment: Small emitter cluster and receivers both with ERK-KTR-YFP with consecutive bPAC input pulses of the same amplitude $F_{max}$. Overshoot in emitter ERK-KTR N/C signals can easily be seen in nuclei 19, 26, 43, and 47. Mobility is observed for a group of receivers above the small emitter cluster.
(AVI)

**S4 Movie. Description: Movie of single emitter cluster.** Details of experiment: Single emitter and receivers both with ERK-KTR-YFP with consecutive bPAC inputs of amplitude $F_{max}$. Overshoot in emitter ERK-KTR N/C signal (nucleus 28) is much more pronounced during second bPAC input pulse. Small group of emitter can be seen on the lower right hand corner, which are more than two receiver cells away from the single emitter.
(AVI)

**S5 Movie. Description: Movie of single emitter clusters that induce strong mobility phenotype.** Details of experiment: Movie of a few emitters induced by consecutive bPAC input pulses of amplitude $F_{max}$. They are surrounded by receivers with over expressed connexin43 with a GFP tagged at its N-terminus which form broken gap-junctions. A group of receivers in the lower half of the field of view exhibit strong mobility about half-way through the experiment.
(AVI)

**S1 Table. The effect of chemical and genetic perturbations on the mobility phenotype.** Table of observations associated with the mobility phenotype under different pharmacological and genetic (emitter or receiver) perturbations. The table is referenced in the Discussion section with deeper discussion in S5 Text.
(PDF)

**S2 Table. Simulation parameter values used for results in Fig 5.** Table of parameters used for the multicellular simulation results presented in Fig 5.
(PDF)

**S3 Table. Simulation parameter values used for results in S8 Fig and Fig 6.** Table of parameters used for the multicellular simulation results presented in S8 Fig and Fig 6.
(PDF)

**S1 Text. Exploring the operational regimes of bPAC and the ERK-KTR reporter.** This section further explores the operational regimes of bPAC and the ERK-KTR reporter that were presented in Fig 2 and S1 Fig. It also further discusses the choice of the ERK-KTR reporter as a downstream reporter of cAMP/PKA activity.
(PDF)

**S2 Text. Variability analysis of ERK-KTR N/C signal across emitters for individual all-emitter experiments.** This section goes through the statistical analysis of single-cell ERK-KTR N/C signals across a population of emitters in an all-emitter monolayer. Specific results were presented in Fig 2 and S2 Fig.
(PDF)

**S3 Text. Both PKA and Epac contribute to the ERK-KTR N/C signal.** This section discusses bPAC experiments with PKA and/or Epac inhibited and the resulting ERK-KTR dynamics.
(PDF)

**S4 Text. Details of multicellular model.** This section goes through the assumptions, parameters and equations of the multicellular model. It also justifies the well-mixed assumption used.
(PDF)

**S5 Text. bPAC induced cAMP in small emitter clusters is causal to a mobility phenotype in receivers.** This section discusses the mobility phenotype under different pharmacological and genetic (emitter or receiver) perturbations. It goes into a more detailed and deeper discussion than the summary at the end of the Discussion section.
(PDF)

## Acknowledgments

We would like to thank Zara Weinberg, Nina Riehs, Lindsey Osimiri, and Jacob Stewart-Ornstein for fruitful discussions.

## Author Contributions

**Conceptualization:** João Pedro Fonseca, Michael Chevalier.

**Formal analysis:** João Pedro Fonseca, Andrew H. Ng, Michael Chevalier.

**Funding acquisition:** Michael Chevalier.

**Investigation:** João Pedro Fonseca, Michael Chevalier.

**Resources:** Elham Aslankoohi.

**Software:** Andrew H. Ng, Michael Chevalier.

**Supervision:** Michael Chevalier.

**Writing – original draft:** João Pedro Fonseca, Michael Chevalier.

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
