## [Decision Letter · Decision Letter 0]

30 Aug 2021

Dear Dr. Chevalier,

Thank you very much for submitting your manuscript "Analysis of localized cAMP perturbations within a tissue reveal the effects of a local, dynamic gap junction state on ERK signaling" for consideration at PLOS Computational Biology.

As with all papers reviewed by the journal, your manuscript was reviewed by members of the editorial board and by several independent reviewers. In light of the reviews (below this email), we would like to invite the resubmission of a significantly-revised version that takes into account the reviewers' comments.

We cannot make any decision about publication until we have seen the revised manuscript and your response to the reviewers' comments. Your revised manuscript is also likely to be sent to reviewers for further evaluation.

Sincerely,

Martin Meier-Schellersheim

Associate Editor

PLOS Computational Biology

Jason Haugh

Deputy Editor

PLOS Computational Biology

Reviewer's Responses to Questions

**Comments to the Authors:**

Reviewer #1: Review Comments to the Author

In the present work entitled: “Analysis of localized cAMP perturbations within a tissue reveal the effects of a local, dynamic gap junction state on ERK signaling”, Fonseca J. and colleagues showed by a combination of pharmacological, imaging and computational modeling approaches that cAMP production crosses gap junction in epithelial cells to inhibit ERK signaling. Authors performed complementary experiments at cellular level using pertinent emitter cell model of bPAC with receiver cells expressing signaling pathway reporters. First, this study, describes their study models and relationships between bPAC induced emitters (in small and single emitter cluster) with ERK-KTR in emitters and receivers. Next, authors performed empirical studies to fit their experimental observations with an appropriate computational model.

Overall, we found this study to be well written although the topic and conclusion might seem narrowed. Some experiments and associated conclusions are clearly convincing, some, however, need additional controls or re-analysis. Therefore, we have comments to address to authors, which would strengthen the paper and conclusions.

Major comments:

#1. Authors provided an extensive work here with a large amount of analysis. However, this large amount of information often makes the work difficult to follow. Therefore, the rational between experiment/data are often confused and unclear that would need explanation/introduction. Furthermore, some supplementary data do not provide additional information to the story and should be removed. Also (but not only), Fig S5 looks to us more pertinent to show than some panels in Fig 4 (ie PKA and Epac). Therefore, we recommend to authors to reorganize the paper (text and figure) to optimize their message (somewhat important) and increase readability.

#2. We believe that some additional control would support their conclusion. Hence, does authors would observe similar data after uncaging cAMP (instead of dPAC)? This would support their observation and validate somehow the proposed mathematical model.

Does the illumination used to stimulate cAMP production affects ERK-KTR in receiver cells lacking bPAC? Does blue pulse affects directly gap junction communication?

#3. In all figures, authors exhibit traces of single cells, we regret that they do not provide more amalgamed analysis with appropriate statistics (eg time to peak, time to return to steady state, decay …). Also, in Fig 3 and 4, we would like to see pictures of cell status at several pertinent time points.

#4. Considering cellular and intercellular cAMP diffusion velocity but also cellular resistivity between connected cells, authors should report a delay in the activation and inhibition of ERK-KTR N/C ratio between emitter and receiver at the beginning and at the end of bPAC stimulation with blue light pulse but also in the intensity of N/C ratio. Therefore, is there any distance effect in N/C intensity, delay in N/C activation/inhibition between proximal and distal receiver cells from emitter cells and a relationship between number of connected cells (cellular resistivity effect) between emitters and connected receivers. These should be mentioned.

Furthermore, authors should discuss their data regarding the literature reporting cAMP diffusion in cytoplasm and between cells (eg Bock A et al Agarwal SH et al and Ponsioen B et al).

Additional comments:

#a. In fig2 legend. There is a switch between panels F and G that should be E and F.

#b. In page 11, the sentence ‘For this case, the cAMP degradation term is simply …. Multiplied BY the rate …’ and not bay.

#c. Authors mentioned the used of carbonoxolone in their study, we guess it was carbenoxolone (CBX).

#d. In Fig S3 legend. Panel C, authors mentioned Top and bottom, instead of left and right

Reviewer #2: I read this paper with great interest and found both the experimental details and the questions fascinating. Since I'm not an experimentalist, I will leave it to the other reviewers to comment on the various controls. However, I have a fundamental question -- what is the purpose of the modeling? I don't mean this to be disrespectful of the authors efforts but I think the text doesn't clarify why a model is necessary and what is to be gained by this additional effort. Since the experiments are so well-contained and the manuscript as written doesn't necessarily test any predictions that were made from the model, it could be interpreted as an exercise in curve-fitting (to be clear, I'm not saying that it is). I'm wondering if there is a missed opportunity here for the authors to make the case that the model clarifies something that is not experimentally obvious or feasible.

Some more detailed points:

1. Time dependent signal varies from cell to cell -- but do you separate signals that arise from gap junction-mediated cAMP versus cell made cAMP?

2. What about growth hormones from the culture medium? What about the naturally present cyclase activation? Are these not important?

3. Time scale of the stimulus? Seems rather long -- 40 min? Why was this chosen? What are the implications of this timescales of cellular health?

4. There is no figure 2G marked.

5. How are the ratios of emitters to receivers controlled? Is it possible to control them?

6. Figure 3: Hard to make the case for any strong correlation between single cell emitters and their neighbors. Is there a critical number of cells needed for a cluster to be functional?

7. Model details: So I appreciated the simplicity of the model but as noted above, it is not clear what purpose was served. Some considerations that should be considered in this case:

a. If each cell is a well-mixed compartment, then why not consider a multicompartment model and use the gap junctions as flux between comparements? This would be the most logical tradeoff between a detailed spatial model and the current version of the model.

b. What about the role of noise? Noise would be inherent in these cellular phenomena and the role of extracellular and intracellular noise should be discussed and somehow incorporated in the models.

c. I think given the imaging that the authors have a rudimentary spatial model would allow for more insight into the diffusion of signals (after all, the gap junctions are boundary fluxes then) rather than the current formulation.

Stylistic: intro can be shortened to focus on the main problem. (This is obviously each person's preference but I'm sharing in case it is useful)

Minor: Check formatting of quotation signs and subscripts for calcium, ip3 etc.

The use of the word tissue is misleading perhaps. Suggest that the authors stick with monolayers?

**Have the authors made all data and (if applicable) computational code underlying the findings in their manuscript fully available?**

Reviewer #1: Yes

Reviewer #2: Yes

PLOS authors have the option to publish the peer review history of their article (what does this mean?). If published, this will include your full peer review and any attached files.

Reviewer #1: **Yes: **Guillaume PIDOUX, INSERM CRCN, PhD

Reviewer #2: No
---

## [Decision Letter · Decision Letter 1]

27 Jan 2022

Dear Dr. Chevalier,

We are pleased to inform you that your manuscript 'Analysis of localized cAMP perturbations within a tissue reveal the effects of a local, dynamic gap junction state on ERK signaling' has been provisionally accepted for publication in PLOS Computational Biology.

Best regards,

Martin Meier-Schellersheim

Associate Editor

PLOS Computational Biology

Jason Haugh

Deputy Editor

PLOS Computational Biology

Reviewer's Responses to Questions

**Comments to the Authors:**

Reviewer #1: Authors provide a revised manuscript of their study: Analysis of localized cAMP perturbations within a tissue reveal the effects of local, dynamic gap junction state on ERK signaling. Authors addressed all our comments which clearly strengthen the paper.

We believe this new version ready to be publish.

Reviewer #2: The authors addressed all my comments in the revised version. I particularly appreciated the clarity of the model and its application in this version. I have no further comments.

**Have the authors made all data and (if applicable) computational code underlying the findings in their manuscript fully available?**

Reviewer #1: Yes

Reviewer #2: Yes

PLOS authors have the option to publish the peer review history of their article (what does this mean?). If published, this will include your full peer review and any attached files.

Reviewer #1: **Yes: **Guillaume Pidoux

Reviewer #2: No

---

## [Editor Report · Acceptance letter]

23 Mar 2022

PCOMPBIOL-D-21-01143R1 

Analysis of localized cAMP perturbations within a tissue reveal the effects of a local, dynamic gap junction state on ERK signaling

Dear Dr Chevalier,

I am pleased to inform you that your manuscript has been formally accepted for publication in PLOS Computational Biology. Your manuscript is now with our production department and you will be notified of the publication date in due course.

With kind regards,

Livia Horvath
